# A Probabilistic Graph Coupling View of Dimension Reduction

**Hugues Van Assel**
UMPA
ENS Lyon
hugues.van_assel@ens-lyon.fr

**Thibault Espinasse**
Institut Camille Jordan Lyon 1
Inria Dracula
espinasse@math.univ-lyon1.fr

**Julien Chiquet**
AgroParisTech
INRAE
julien.chiquet@inrae.fr

**Franck Picard**
LBMC, CNRS
ENS Lyon
franck.picard@ens-lyon.fr

## Abstract

Most popular dimension reduction (DR) methods like t-SNE and UMAP are based on minimizing a cost between input and latent pairwise similarities. Though widely used, these approaches lack clear probabilistic foundations to enable a full understanding of their properties and limitations. To that extent, we introduce a unifying statistical framework based on the coupling of hidden graphs using cross entropy. These graphs induce a Markov random field dependency structure among the observations in both input and latent spaces. We show that existing pairwise similarity DR methods can be retrieved from our framework with particular choices of priors for the graphs. Moreover this reveals that these methods relying on shift-invariant kernels suffer from a statistical degeneracy that explains poor performances in conserving coarse-grain dependencies. New links are drawn with PCA which appears as a non-degenerate graph coupling model.

## 1 Introduction

Dimensionality reduction (DR) is of central importance when dealing with high-dimensional data [15]. It mitigates the curse of dimensionality, allowing for greater statistical flexibility and less computational complexity. DR also enables visualization that can be of great practical interest for understanding and interpreting the structure of large datasets. Most seminal approaches include Principal Component Analysis (PCA) [31], multidimensional scaling [23] and more broadly kernel eigenmaps methods such as Isomap [4], Laplacian eigenmaps [5] and diffusion maps [12]. These methods share the definition of a pairwise similarity kernel that assigns a high value to close neighbors and the resolution of a spectral problem. They are well understood and unified in the kernel PCA framework [16].

In the past decade, the field has witnessed a major shift with the emergence of a new class of methods. They are also based on pairwise similarities but these are not converted into inner products. Instead, they define pairwise similarity functions in both input and latent spaces and optimize a cost between the two. Among such methods, the Stochastic Neighbor Embedding (SNE) algorithm [18], its heavy-tailed symmetrized version t-SNE [38] or more recent approaches like LargeVis [36] and UMAP [30] are arguably the most used in practice. These will be referred to as *SNE-like* or *neighbor embedding* methods in what follows. They are increasingly popular and now considered as the state-of-art techniques in many fields

36th Conference on Neural Information Processing Systems (NeurIPS 2022).

[27, 19, 1]. Their popularity is mainly due to their exceptional ability to preserve local structure, *i.e.* close points in the input space have close embeddings, as shown empirically [40]. They also demonstrate impressive performances in identifying clusters [3, 29]. However this is done at the expense of global structure, that these methods struggle in preserving [41, 11] *i.e.* the relative large-scale distances between embedded points do not necessarily correspond to the original ones.

Due to a lack of clear probabilistic foundations, these properties remain mostly empirical. This gap between theory and practice is detrimental as practitioners may rely on strategies that are not optimal for their use case. While recent software developments are making these methods more scalable [8, 33, 28] and further expanding their use, the need for a well-established probabilistic framework is becoming more prominent. In this work we define the generative probabilistic model that encompasses current embedding methods, while establishing new links with the well-established PCA model.

**Outline.** Consider $\boldsymbol{X} = (\boldsymbol{X}_1, ..., \boldsymbol{X}_n)^\top \in \mathbb{R}^{n \times p}$, an input dataset that consists of $n$ vectors of dimension $p$. Our task is to embed $\boldsymbol{X}$ in a lower dimensional space of dimension $q < p$ (typically $q = 2$ for visualization), and we denote by $\boldsymbol{Z} = (\boldsymbol{Z}_1, ..., \boldsymbol{Z}_n)^\top \in \mathbb{R}^{n \times q}$ the unknown embeddings. The rationale of our framework is to suppose that the observations $\boldsymbol{X}$ and $\boldsymbol{Z}$ are structured by two latent graphs with $\boldsymbol{W}_X$ and $\boldsymbol{W}_Z$ standing for their $n$-square weight matrices. As the goal of DR is to preserve the input's structure in the latent space, we propose to find the best low-dimensional representation $\boldsymbol{Z}$ of $\boldsymbol{X}$ such that $\boldsymbol{W}_X$ and $\boldsymbol{W}_Z$ are close. To build a flexible and robust probabilistic framework, we consider random graphs distributed according to some predefined prior distributions. Our objective is to match the posterior distributions of $\boldsymbol{W}_X$ and $\boldsymbol{W}_Z$. Note that as they share the same dimensionality the latter graphs can be easily compared unlike $\boldsymbol{X}$ and $\boldsymbol{Z}$. The coupling is done with a cross entropy criterion, the minimization of which will be referred to as graph coupling.

In this work, our main contributions are as follows.

- We show that SNE, t-SNE, LargeVis and UMAP are all instances of graph coupling and characterized by different choices of prior for discrete latent structuring graphs (section 3). We demonstrate that such graphs essentially capture conditional independencies among rows through a pairwise Markov Random Field (MRF) model which construction can be found in section 2.

- We uncover the intrinsic probabilistic property explaining why such methods perform poorly on conserving the large scale structure of the data as a consequence of a degeneracy of the MRF when shift invariant kernels are used (theorem 1). Such degeneracy induces the loss of the relative positions of clusters corresponding to the connected components of the posterior latent graphs which distributions are identified (proposition 1). These findings are highlighted by a new initialization of the embeddings that is empirically tested (section 4).

- We show that for Gaussian MRFs, when adapting graph coupling to precision matrices with suitable priors, PCA appears as a natural extension of the coupling problem in its continuous version (theorem 2). Such model does not suffer from the aforementioned degeneracy hence preserves the large-scale structure.

## 2  Shift-Invariant Pairwise MRF to Model Row Dependencies

We start by defining the distribution of the observations given a graph. The latter takes the form of a pairwise MRF model which as we show is improper (*i.e.* not integrable on $\mathbb{R}^{n \times p}$) when shift-invariant kernels are used. We consider a fixed directed graph $\boldsymbol{W} \in \mathcal{S}_W$ where:

$$\mathcal{S}_W = \left\{ \boldsymbol{W} \in \mathbb{N}^{n \times n} \mid \forall (i, j) \in [n]^2, W_{ii} = 0, W_{ij} \leq n \right\}$$

Throughout, $(E, \mathcal{B}(E), \lambda_E)$ denotes a measure space where $\mathcal{B}(E)$ is the Borel $\sigma$-algebra on $E$ and $\lambda_E$ is the Lebesgue measure on $E$.

## 2.1 Graph Laplacian Null Space

A central element in our construction is the graph Laplacian linear map, defined as follows, where $\mathcal{S}_+^n(\mathbb{R})$ is the set of positive semidefinite matrices.

**Definition 1** *The graph Laplacian operator is the map* $L\colon \mathbb{R}_+^{n\times n} \to \mathcal{S}_+^n(\mathbb{R})$ *such that*

$$for\ (i,j) \in [n]^2, \quad L(\boldsymbol{W})_{ij} = \begin{cases} -W_{ij} & if\ i \neq j \\ \sum_{k\in[n]} W_{ik} & otherwise\,. \end{cases}$$

With an abuse of notation, let $\boldsymbol{L} = L(\overline{\boldsymbol{W}})$ where $\overline{\boldsymbol{W}} = \boldsymbol{W} + \boldsymbol{W}^\top$. Let $(C_1, ..., C_R)$ be a partition of $[n]$ (*i.e.* the set $\{1, 2, ..., n\}$) corresponding to the connected components (CCs) of $\overline{\boldsymbol{W}}$. As well known in spectral graph theory [9], the null space of $\boldsymbol{L}$ is spanned by the orthonormal vectors $\{\boldsymbol{U}_r\}_{r\in[R]}$ such that for $r \in [R]$, $\boldsymbol{U}_r = \left(n_r^{-1/2}\mathbb{1}_{i\in C_r}\right)_{i\in[n]}$ with $n_r = \mathrm{Card}(C_r)$. By the spectral theorem, $\boldsymbol{U}_{[R]}$ can be completed such that $\boldsymbol{L} = \boldsymbol{U}\boldsymbol{\Lambda}\boldsymbol{U}^\top$ where $\boldsymbol{U} = (\boldsymbol{U}_1, ..., \boldsymbol{U}_n)$ is orthogonal and $\boldsymbol{\Lambda} = \mathrm{diag}((\lambda_i)_{i\in[n]})$ with $0 = \lambda_1 = ... = \lambda_R < \lambda_{R+1} \leq ... \leq \lambda_n$. In the following, the data is split into two parts: $\boldsymbol{X}_M$, the orthogonal projection of $\boldsymbol{X}$ on $\mathcal{S}_M = (\ker \boldsymbol{L}) \otimes \mathbb{R}^p$, and $\boldsymbol{X}_C$, the projection on $\mathcal{S}_C = (\ker \boldsymbol{L})^\perp \otimes \mathbb{R}^p$. For $i \in [n]$, $\boldsymbol{X}_{M,i} = \sum_{r\in[R]} n_r^{-1} \mathbb{1}_{i\in C_r} \sum_{\ell\in C_r} \boldsymbol{X}_\ell$ hence $\boldsymbol{X}_M$ stands for the empirical means of $\boldsymbol{X}$ on CCs, thus modelling the CC positions, while $\boldsymbol{X}_C = \boldsymbol{X} - \boldsymbol{X}_M$ is CC-wise centered, thus modeling the relative positions of the nodes within CCs. We now introduce the probability distribution of these variables.

## 2.2 Pairwise MRF and Shift-Invariances

In this work, the dependency structure among rows of the data is governed by a graph. The strength of the connection between two nodes is given by a symmetric function $k : \mathbb{R}^p \to \mathbb{R}_+$. We consider the following pairwise MRF unnormalized density function:

$$f_k\colon (\boldsymbol{X}, \boldsymbol{W}) \mapsto \prod_{(i,j)\in[n]^2} k(\boldsymbol{X}_i - \boldsymbol{X}_j)^{W_{ij}}\,. \tag{1}$$

As we will see shortly, the above is at the heart of DR methods based on pairwise similarities. Note that as $k$ measures the similarity between couples of samples, $f_k$ will take high values if the rows of $\boldsymbol{X}$ vary smoothly on the graph $\boldsymbol{W}$. Thus we can expect $\boldsymbol{X}_i$ and $\boldsymbol{X}_j$ to be close if there is an edge between node $i$ and node $j$ in $\boldsymbol{W}$. A key remark is that $f_k$ is kept invariant by translating $\boldsymbol{X}_M$. Namely for all $\boldsymbol{X} \in \mathbb{R}^{n\times p}$, $f_k(\boldsymbol{X}, \boldsymbol{W}) = f_k(\boldsymbol{X}_C, \boldsymbol{W})$. This invariance results in $f_k(\cdot, \boldsymbol{W})$ being non integrable on $\mathbb{R}^{n\times p}$, as we see with the following example.

**Gaussian kernel.** For a positive definite matrix $\boldsymbol{\Sigma} \in \mathcal{S}_{++}^n(\mathbb{R})$, consider the Gaussian kernel $k : \boldsymbol{x} \mapsto e^{-\frac{1}{2}\|\boldsymbol{x}\|_{\boldsymbol{\Sigma}}^2}$ where $\boldsymbol{\Sigma}$ stands for the covariance among columns. One has:

$$\log f_k(\boldsymbol{X}, \boldsymbol{W}) = -\sum_{(i,j)\in[n]^2} W_{ij}\|\boldsymbol{X}_i - \boldsymbol{X}_j\|_{\boldsymbol{\Sigma}}^2 = -\mathrm{tr}\left(\boldsymbol{\Sigma}^{-1}\boldsymbol{X}^T\boldsymbol{L}\boldsymbol{X}\right) \tag{2}$$

by property of the graph Laplacian (definition 1). In this case, it is clear that due to the rank deficiency of $\boldsymbol{L}$, $f_k(\cdot, \boldsymbol{W})$ is only $\lambda_{\mathcal{S}_C}$-integrable. In general DR settings one does not want to rely on Gaussian kernels only. A striking example is the use of the Student kernel in t-SNE [38]. Heavy-tailed kernels appear useful when the dimension of the embeddings is smaller than the intrinsic dimension of the data [20]. Our contribution provides flexibility by extending the previous result to a large class of kernels, as stated in the following theorem.

**Theorem 1** *If $k$ is $\lambda_{\mathbb{R}^p}$-integrable and bounded above $\lambda_{\mathbb{R}^p}$-almost everywhere then $f_k(\cdot, \boldsymbol{W})$ is $\lambda_{\mathcal{S}_C}$-integrable.*

We refer to appendix A.1 for the proof. We can now define a distribution on $(\mathcal{S}_C, \mathcal{B}(\mathcal{S}_C))$, where $\mathcal{C}_k(\boldsymbol{W}) = \int f_k(\cdot, \boldsymbol{W})d\lambda_{\mathcal{S}_C}$:

$$\mathbb{P}_k(d\boldsymbol{X}_C|\boldsymbol{W}) = \mathcal{C}_k(\boldsymbol{W})^{-1}f_k(\boldsymbol{X}_C, \boldsymbol{W})\lambda_{\mathcal{S}_C}(d\boldsymbol{X}_C)\,. \tag{3}$$

**Remark 1** *Kernels may have node-specific bandwidths $\boldsymbol{\tau}$, set during a pre-processing step, giving $f_k(\boldsymbol{X}, \boldsymbol{W}) = \prod_{(i,j)} k((\boldsymbol{X}_i - \boldsymbol{X}_j)/\tau_i)^{W_{ij}}$. Note that such bandwidth does not affect the degeneracy of the distribution and theorem 1 still holds.*

**Between-Rows Dependency Structure.** By symmetry of $k$, reindexing gives: $f_k(\boldsymbol{X}, \boldsymbol{W}) = \prod_{j \in [n]} \prod_{i \in [j]} k(\boldsymbol{X}_i - \boldsymbol{X}_j)^{\overline{W}_{ij}}$. Hence distribution (3) boils down to a pairwise MRF model [10] with respect to the undirected graph $\overline{\boldsymbol{W}}$, $\mathcal{C}_k$ playing the role of the partition function. Note that since $f_k$ (Equation 1) trivially factorize according to the cliques of $\overline{\boldsymbol{W}}$, the Hammersley-Clifford theorem ensures that the rows of $\boldsymbol{X}_C$ satisfy the local and global Markov properties with respect to $\overline{\boldsymbol{W}}$.

### 2.3 Uninformative Model for CC-wise Means

We showed that the MRF (1) is only integrable on $\mathcal{S}_C$, the definition of which depends on the connectivity structure of $\boldsymbol{W}$. As we now demonstrate, the latter MRF can be seen as a limit of proper distributions on $\mathbb{R}^{n \times p}$, see *e.g.* [35] for a similar construction in the Gaussian case. We introduce the Borel function $f^\varepsilon(\cdot, \boldsymbol{W}): \mathbb{R}^{n \times p} \to \mathbb{R}_+$ for $\varepsilon > 0$ such that for all $\boldsymbol{X} \in \mathbb{R}^{n \times p}$, $f^\varepsilon(\boldsymbol{X}, \boldsymbol{W}) = f^\varepsilon(\boldsymbol{X}_M, \boldsymbol{W})$. To allow $f^\varepsilon$ to become arbitrarily non-informative, we assume that for all $\boldsymbol{W} \in \mathcal{S}_W$, $f^\varepsilon(\cdot, \boldsymbol{W})$ is $\lambda_{\mathcal{S}_M}$-integrable for all $\varepsilon \in \mathbb{R}_+^*$ and $f^\varepsilon(\cdot, \boldsymbol{W}) \xrightarrow[\varepsilon \to 0]{} 1$ almost everywhere. We now define the conditional distribution on $(\mathcal{S}_M, \mathcal{B}(\mathcal{S}_M))$ as follows:

$$\mathbb{P}^\varepsilon(d\boldsymbol{X}_M | \boldsymbol{W}) = \mathcal{C}^\varepsilon(\boldsymbol{W})^{-1} f^\varepsilon(\boldsymbol{X}_M, \boldsymbol{W}) \lambda_{\mathcal{S}_M}(d\boldsymbol{X}_M) \tag{4}$$

where $\mathcal{C}^\varepsilon(\boldsymbol{W}) = \int f^\varepsilon(\cdot, \boldsymbol{W}) d\lambda_{\mathcal{S}_M}$. With this at hand, the joint conditional is defined as the product measure of (3) and (4) over the row axis, the integrability of which is ensured by the Fubini-Tonelli theorem. In the following we will use the compact notation $\mathcal{C}_k^\varepsilon(\boldsymbol{W}) = \mathcal{C}_k(\boldsymbol{W})\mathcal{C}^\varepsilon(\boldsymbol{W})$ for the joint normalizing constant.

**Remark 2** *At the limit $\varepsilon \to 0$ the above construction amounts to setting an infinite variance on the distribution of the empirical means of $\boldsymbol{X}$ on CCs, thus loosing the inter-CC structure.*

As an illustration, one can structure the CCs' relative positions according to a Gaussian model with positive definite precision $\varepsilon\boldsymbol{\Theta} \in \mathcal{S}_{++}^R(\mathbb{R})$, as it amounts to choosing $f^\varepsilon: \boldsymbol{X} \to \exp\left(-\frac{\varepsilon}{2} \operatorname{tr}\left(\boldsymbol{\Sigma}^{-1} \boldsymbol{X}^\top \boldsymbol{U}_{[:R]} \boldsymbol{\Theta} \boldsymbol{U}_{[R]}^\top \boldsymbol{X}\right)\right)$ such that: $\operatorname{vec}(\boldsymbol{X}_M) | \boldsymbol{\Theta} \sim \mathcal{N}\left(\boldsymbol{0}, \left(\varepsilon\boldsymbol{U}_{[:R]} \boldsymbol{\Theta} \boldsymbol{U}_{[R]}^\top\right)^{-1} \otimes \boldsymbol{\Sigma}\right)$ where $\otimes$ denotes the Kronecker product.

## 3 Graph Coupling as a Unified Objective for Pairwise Similarity Methods

In this section, we show that neighbor embedding methods can be recovered in the presented framework. They are obtained, for particular choices of graph priors, at the limit $\varepsilon \to 0$ when $f^\varepsilon$ becomes non informative and the CCs' relative positions are lost.

We now turn to the priors for $\boldsymbol{W}$. Our methodology is similar to that of constructing conjugate priors for distributions in the exponential family [39], notably we insert the cumulant function $\mathcal{C}_k^\varepsilon$ (*i.e.* normalizing constant of the conditional) as a multivariate term of the prior. We consider different forms: binary ($B$), unitary out-degree ($D$) and $n$-edges ($E$), relying on an additional term ($\Omega$) to constraint the topology of the graph. For a matrix $\boldsymbol{A}$, $A_{i+}$ denotes $\sum_j A_{ij}$ and $A_{++}$ denotes $\sum_{ij} A_{ij}$. In the following, $\boldsymbol{\pi}$ plays the role of the edge's prior. The latter can be leveraged to incorporate some additional information about the dependency structure, for instance when a network is observed [26].

**Definition 2** *Let $\boldsymbol{\pi} \in \mathbb{R}_+^{n \times n}$, $\varepsilon \in \mathbb{R}_+$, $\alpha \in \mathbb{R}$, $k$ satisfies the assumptions of theorem 1 and $\mathcal{P} \in \{B, D, E\}$. For $\boldsymbol{W} \in \mathcal{S}_W$ we introduce:*

$$\mathbb{P}_{\mathcal{P},k}^\varepsilon(\boldsymbol{W}; \boldsymbol{\pi}, \alpha) \propto \mathcal{C}_k^\varepsilon(\boldsymbol{W})^\alpha \, \Omega_{\mathcal{P}}(\boldsymbol{W}) \prod_{(i,j) \in [n]^2} \pi_{ij}^{W_{ij}}$$

*where $\Omega_B(\boldsymbol{W}) = \prod_{ij} \mathbb{1}_{W_{ij} \leq 1}$, $\Omega_D(\boldsymbol{W}) = \prod_i \mathbb{1}_{W_{i+}=1}$ and $\Omega_E(\boldsymbol{W}) = \mathbb{1}_{W_{++}=n} \prod_{ij} (W_{ij}!)^{-1}$.*

When $\alpha = 0$, the above no longer depends on $\varepsilon$ and $k$. We will use the compact notation $\mathbb{P}_{\mathcal{P}}(\boldsymbol{W}; \boldsymbol{\pi}) = \mathbb{P}_{\mathcal{P},k}^{\varepsilon}(\boldsymbol{W}; \boldsymbol{\pi}, 0)$. Note that by $\boldsymbol{W} \sim \mathbb{P}_{\mathcal{P}}(\cdot\,; \boldsymbol{\pi})$ we have the following simple Bernoulli ($\mathcal{B}$) and multinomial ($\mathcal{M}$) distributions, where matrix or vector division is to be understood as element-wise.

- If $\mathcal{P} = B$, $\forall (i,j) \in [n]^2$, $W_{ij} \overset{\perp\!\!\!\perp}{\sim} \mathcal{B}(\pi_{ij}/(1+\pi_{ij}))$.
- If $\mathcal{P} = D$, $\forall i \in [n]$, $\boldsymbol{W}_i \overset{\perp\!\!\!\perp}{\sim} \mathcal{M}(1, \boldsymbol{\pi}_i/\pi_{i+})$.
- If $\mathcal{P} = E$, $\boldsymbol{W} \sim \mathcal{M}(n, \boldsymbol{\pi}/\pi_{++})$.

We now show that the posterior distribution of the graph given the observations takes a simple form when the distribution of CC empirical means $\boldsymbol{X}_M$ diffuses *i.e.* when $\varepsilon \to 0$ (a proof of the following result can be found in appendix A.2). In the following, $\odot$ stands for the Hadamard product and $\mathcal{D}$ for the convergence in distribution.

**Proposition 1** *Let $\boldsymbol{\pi} \in \mathbb{R}_+^{n\times n}$, $k$ satisfies the assumptions of theorem 1 with $\boldsymbol{K}_X = (k(\boldsymbol{X}_i - \boldsymbol{X}_j))_{(i,j)\in[n]^2}$ and $\mathcal{P} \in \{B, D, E\}$. If $\boldsymbol{W}^{\varepsilon} \sim \mathbb{P}_{\mathcal{P},k}^{\varepsilon}(\cdot\,; \boldsymbol{\pi}, 1)$ then*

$$\boldsymbol{W}^{\varepsilon}|\boldsymbol{X} \xrightarrow[\varepsilon \to 0]{\mathcal{D}} \mathbb{P}_{\mathcal{P}}(\cdot\,; \boldsymbol{\pi} \odot \boldsymbol{K}_X).$$

**Remark 3** *For all $\boldsymbol{W} \in \mathcal{S}_W$, $\mathcal{C}^{\varepsilon}(\boldsymbol{W})$ diverges as $\varepsilon \to 0$, hence the graph prior (definition 2) is improper at the limit. This compensates for the uninformative diffuse conditional and allows to retrieve a well-defined tractable posterior limit.*

### 3.1 Retrieving Well Known DR Methods

We now provide a unified view of neighbor embedding objectives as a coupling between graph posterior distributions. To that extent we derive the cross entropy associated with the various graph priors at hand. In what follows, $k_x$ and $k_z$ satisfy the assumptions of theorem 1 and we denote by $\boldsymbol{K}_X$ and $\boldsymbol{K}_Z$ the associated kernel matrices on $\boldsymbol{X}$ and $\boldsymbol{Z}$ respectively. For both graph priors we consider the parameters $\boldsymbol{\pi} = \boldsymbol{1}$ and $\alpha = 1$. For $(\mathcal{P}_X, \mathcal{P}_Z) \in \{B, D, E\}^2$, we introduce the cross entropy between the limit posteriors at $\varepsilon \to 0$,

$$\mathcal{H}_{\mathcal{P}_x, \mathcal{P}_z} = -\mathbb{E}_{\boldsymbol{W}_X \sim \mathbb{P}_{\mathcal{P}_x}(\cdot\,; \boldsymbol{K}_X)}[\log \mathbb{P}_{\mathcal{P}_z}(\boldsymbol{W}_Z = \boldsymbol{W}_X; \boldsymbol{K}_Z)]$$

defining a coupling criterion to be optimized with respect to embedding coordinates $\boldsymbol{Z}$. We now go through each couple $(\mathcal{P}_X, \mathcal{P}_Z)$ such that $\mathrm{supp}(\mathbb{P}_{\mathcal{P}_x}) \subset \mathrm{supp}(\mathbb{P}_{\mathcal{P}_z})$ for the cross-entropy to be defined.

**SNE.** When $\mathcal{P}_X = \mathcal{P}_Z = D$, the probability of the limit posterior graphs factorizes over the nodes and the cross-entropy between limit posteriors takes the form of the objective of SNE [17], where for $i \in [n]$, $\boldsymbol{P}_i^D = \boldsymbol{K}_{X,i}/K_{X,i+}$ and $\boldsymbol{Q}_i^D = \boldsymbol{K}_{Z,i}/K_{Z,i+}$,

$$\mathcal{H}_{D,D} = -\sum_{i\neq j} P_{ij}^D \log Q_{ij}^D.$$

**Symmetric-SNE.** Choosing $\mathcal{P}_X = D$ and $\mathcal{P}_Z = E$, we define for $(i,j) \in [n]^2$, $\boldsymbol{Q}_{ij}^E = K_{Z,ij}/K_{Z,++}$ and $\overline{P}_{ij}^D = P_{ij}^D + P_{ji}^D$. The symmetry of $\boldsymbol{Q}^E$ yields:

$$\mathcal{H}_{D,E} = -\sum_{i\neq j} P_{ij}^D \log Q_{ij}^E = -\sum_{i<j} \overline{P}_{ij}^D \log Q_{ij}^E$$

and the symmetrized objective of t-SNE [38] is recovered.

**LargeVis.** Now choosing $\mathcal{P}_X = D$ and $\mathcal{P}_Z = B$, one can also notice that $\boldsymbol{Q}^B = (K_{Z,ij}/(1+K_{Z,ij}))_{(i,j)\in[n]^2}$ is symmetric. With this at hand the limit cross-entropy reads

$$\mathcal{H}_{D,B} = -\sum_{i\neq j} P_{ij}^D \log Q_{ij}^B + (1 - P_{ij}^D) \log(1 - Q_{ij}^B) = -\sum_{i<j} \overline{P}_{ij}^D \log Q_{ij}^B + \left(2 - \overline{P}_{ij}^D\right) \log(1 - Q_{ij}^B)$$

which is the objective of LargeVis [36].

Table 1: Prior distributions for $\boldsymbol{W}_X$ and $\boldsymbol{W}_Z$ associated with the pairwise similarity coupling DR algorithms. Grey-colored boxes are such that the cross-entropy is undefined.

| $\mathcal{P}_X$ \ $\mathcal{P}_Z$ | $B$ | $D$ | $E$ |
|---|---|---|---|
| $\widetilde{B}$ | UMAP | | |
| $D$ | LARGEVIS | SNE | T-SNE |

**UMAP.** Let us take $\mathcal{P}_X = \mathcal{P}_Z = B$ and consider the symmetric thresholded graph $\widetilde{\boldsymbol{W}}_X = \mathbb{1}_{\boldsymbol{W}_X + \boldsymbol{W}_X^\top \geq 1}$. By independence of the edges, $\widetilde{W}_{X,ij} \sim \mathcal{B}\left(\widetilde{P}_{ij}^B\right)$ where $\widetilde{P}_{ij}^B = P_{ij}^B + P_{ji}^B - P_{ij}^B P_{ji}^B$ and $\boldsymbol{P}^B = (K_{X,ij}/(1 + K_{X,ij}))_{(i,j) \in [n]^2}$. Coupling $\widetilde{\boldsymbol{W}}_X$ and $\boldsymbol{W}_Z$ gives:

$$\mathcal{H}_{\widetilde{B},B} = -2 \sum_{i<j} \widetilde{P}_{ij}^B \log Q_{ij}^B + \left(1 - \widetilde{P}_{ij}^B\right) \log\left(1 - Q_{ij}^B\right)$$

which is the loss function considered in UMAP [30], the construction of $\widetilde{\boldsymbol{W}}_X$ being borrowed from section 3.1 of the paper.

**Remark 4** *One can also consider $\mathcal{H}_{E,E}$ but as detailed in [38], this criterion fails at positioning outliers and is therefore not considered. Interestingly, any other feasible combination of the presented priors relates to an existing method.*

### 3.2 Interpretations

As we have seen in section 3.1, SNE-like methods can all be derived from the graph coupling framework. What characterizes each of them is the choice of priors considered for the latent structuring graphs. To the best of our knowledge, the presented framework is the first that manages to unify all these DR algorithms. Such a framework opens many perspectives for improving upon current practices as we discuss in section 4 and section 5. We now focus on a few insights that our work provides about the empirical performances of these methods.

**Repulsion & Attraction.** Decomposing $\mathcal{H}_{\mathcal{P}_X,\mathcal{P}_Z}$ with Bayes' rule and simplifying constant terms one has the following optimization problem:

$$\min_{\boldsymbol{Z} \in \mathbb{R}^{n \times q}} - \sum_{(i,j) \in [n]^2} \boldsymbol{P}_{ij}^{\mathcal{P}_X} \log k_z(\boldsymbol{Z}_i - \boldsymbol{Z}_j) + \log \mathbb{P}(\boldsymbol{Z}). \qquad (5)$$

The first and second terms in eq. (5) respectively summarize the attractive and repulsive forces of the objective. Recall from proposition 1 that $\boldsymbol{P}^{\mathcal{P}_X}$ is the posterior expectation of $\boldsymbol{W}_X$. Hence in SNE-like methods, the attractive forces resume to a pairwise MRF log likelihood with respect to a graph posterior expectation given $\boldsymbol{X}$. For instance if $k_z$ is the Gaussian kernel, this attractive term reads $\text{tr}\left(\boldsymbol{Z}^\top \boldsymbol{L}^\star \boldsymbol{Z}\right)$ where $\boldsymbol{L}^\star = \mathbb{E}_{\boldsymbol{W} \sim \mathbb{P}_{\mathcal{P}_X}(\cdot;\boldsymbol{K}_X)}[L(\boldsymbol{W})]$, boiling down to the objective of Laplacian eigenmaps [5]. Therefore, for Gaussian MRFs, the attractive forces resume to an unconstrained Laplacian eigenmaps objective. Such link, already noted in [7], is easily unveiled in our framework. Moreover, one can notice that only this attractive term depends on $\boldsymbol{X}$ as the repulsion is given by the marginal term in (5). The latter reads $\mathbb{P}(\boldsymbol{Z}) = \sum_{\boldsymbol{W} \in \mathcal{S}_W} \mathbb{P}(\boldsymbol{Z}, \boldsymbol{W})$ with $\mathbb{P}(\boldsymbol{Z}, \boldsymbol{W}) \propto f_k(\boldsymbol{Z}, \boldsymbol{W}) \Omega_{\mathcal{P}_Z}(\boldsymbol{W})$. Such penalty notably prevents a trivial solution, as $\boldsymbol{0}$, like any constant vector, is a mode of $f_k(\cdot, \boldsymbol{W})$ for all $\boldsymbol{W}$. Also note that the prior for $\boldsymbol{W}_X$ only conditions attraction while the prior for $\boldsymbol{W}_Z$ only affects repulsion. In the present work we focus solely on deciphering the probabilistic model that accounts for neighbor embedding loss functions and refer to [6] for a quantitative study of attraction and repulsion in these methods.

**Global Structure Preservation.** To gain intuition, consider that $\boldsymbol{W}_X$ is observed. As we showed in section 2.2, when one relies on shift invariant kernels, the positions of the CC means are taken from a diffuse distribution. Since the above methods are all derived

from the limit posteriors at $\varepsilon \to 0$, $\boldsymbol{X}_M$ and $\boldsymbol{Z}_M$ have no influence on the coupling objective. Hence if two nodes belong to different CCs, their low dimensional pairwise distance will likely not be faithful. We can expect this phenomenon to persist when the expectation on $\boldsymbol{W}_X$ is considered, especially when clusters are well distinguishable in $\boldsymbol{X}$. This observation is central to understand the large scale deficiency of these methods. Note that this happens at the benefit of the local structure which is faithfully represented in low dimension, as discussed in section 1. In the following section we propose to mitigate the global structure deficiency with non-degenerate MRF models.

## 4 Towards Capturing Large-Scale Dependencies

In this section, we investigate the ability of graph coupling to faithfully represent global structure in low dimension. To gain intuition on the case where the distribution induced by the graph is not degenerate, we consider a proper Gaussian graph coupling model and show its equivalence with PCA. We then provide a new initialization procedure to alleviate the large scale deficiency of graph coupling when degenerate MRFs are used.

### 4.1 PCA as Graph Coupling

As we argue that the inability of SNE-like methods to reproduce the coarse-grain dependencies of the input in the latent space is due to the degeneracy of the conditional (3), a natural solution would be to consider graphical models that are well defined and integrable on the entire definition spaces of $\boldsymbol{X}$ and $\boldsymbol{Z}$. For simplicity, we consider the Gaussian model and leave the extension to other kernels for future works. Note that in this case integrability translates into the precision matrix being full-rank. As we see with the following, the natural extension of our framework to such models leads to a well-established PCA algorithm. In the following, for a continuous variable $\boldsymbol{\Theta}_Z$, $\mathbb{P}(\boldsymbol{\Theta}_Z = \cdot)$ denotes its density.

**Theorem 2** *Let $\nu \geq n$, $\boldsymbol{\Theta}_X \sim \mathcal{W}(\nu, \boldsymbol{I}_n)$ and $\boldsymbol{\Theta}_Z \sim \mathcal{W}(\nu + p - q, \boldsymbol{I}_n)$. Assume that $\boldsymbol{\Theta}_X$ and $\boldsymbol{\Theta}_Z$ structure the rows of respectively $\boldsymbol{X}$ and $\boldsymbol{Z}$ such that:*

$$\text{vec}(\boldsymbol{X})|\boldsymbol{\Theta}_X \sim \mathcal{N}(\boldsymbol{0}, \boldsymbol{\Theta}_X^{-1} \otimes \boldsymbol{I}_p), \tag{6}$$

$$\text{vec}(\boldsymbol{Z})|\boldsymbol{\Theta}_Z \sim \mathcal{N}(\boldsymbol{0}, \boldsymbol{\Theta}_Z^{-1} \otimes \boldsymbol{I}_q). \tag{7}$$

*Then the solution of the precision coupling problem:*

$$\min_{\boldsymbol{Z} \in \mathbb{R}^{n \times q}} -\mathbb{E}_{\boldsymbol{\Theta}_X|\boldsymbol{X}} \left[ \log \mathbb{P}(\boldsymbol{\Theta}_Z = \boldsymbol{\Theta}_X|\boldsymbol{Z}) \right]$$

*is a PCA embedding of $\boldsymbol{X}$ with $q$ components.*

We now highlight the parallels with the previous construction done for neighbor embedding methods. First note that the multivariate Gaussian with full-rank precision is inherently a pairwise MRF [35]. When choosing the Gaussian kernel for neighbor embedding methods, we saw that the graph Laplacian $\boldsymbol{L}_X$ of $\boldsymbol{W}_X$ was playing the role of the among-row precision matrix, as we had $\boldsymbol{X}|\boldsymbol{W}_X \sim \mathcal{N}(\boldsymbol{0}, \boldsymbol{L}_X^{-1} \otimes \boldsymbol{I}_p)$ (equation 2). Recall that the later always has a null-space which is spanned by the CC indicator vectors of $\boldsymbol{W}$ (section 2.1). Here, the key difference is that we impose a full-rank constraint on the precision $\boldsymbol{\Theta}$. Concerning the priors, we choose the ones that are conjugate to the conditionals (6) and (7), as previously done when constructing the prior for neighbor embedding methods (definition 2). Hence in the full-rank setting, the prior simply amounts to a Wishart distribution denoted by $\mathcal{W}$.

The above theorem further highlights the flexibility and generality of the graph coupling framework. Unlike usual constructions of PCA or probabilistic PCA [37], in the above the linear relation between $\boldsymbol{X}$ and $\boldsymbol{Z}$ is recovered by solving the graph coupling problem and not explicitly stated beforehand. To the best of our knowledge, it is the first time such a link is uncovered between PCA and SNE-like methods. In contrast with the latter, PCA is well-known for its ability to preserve global structure while being significantly less efficient at identifying clusters [2]. Therefore, as suspected in section 3.2, the degeneracy of the conditional distribution given the graph is key to determine the distance preservation properties of the embeddings. We propose in section 4.2 to combine both graph coupling approaches to strike a balance between global and local structure preservation.

## 4.2 Hierarchical Graph Coupling

The goal of this section is to show that global structure in SNE-like embeddings can be improved by structuring the CCs' positions. We consider the following hierarchical model for $\boldsymbol{X}$, where $\mathcal{P}_X \in \{B, D, E\}$, $k_x$ satisfies the assumptions of theorem 1 and $\nu_X \geq n$:

$$\boldsymbol{W}_X \sim \mathbb{P}^{\varepsilon}_{\mathcal{P}_X, k_x}(\,\cdot\,; \mathbf{1}, 1), \quad \boldsymbol{\Theta}_X | \boldsymbol{W}_X \sim \mathcal{W}(\nu_X, \boldsymbol{I}_R)$$

$$\boldsymbol{X}_C | \boldsymbol{W}_X \sim \mathbb{P}_{k_x}(\,\cdot\, | \boldsymbol{W}_X), \quad \text{vec}(\boldsymbol{X}_M) | \boldsymbol{\Theta}_X \sim \mathcal{N}\left(\mathbf{0}, \left(\varepsilon \boldsymbol{U}_{[:R]} \boldsymbol{\Theta}_X \boldsymbol{U}_{[:R]}^{\top}\right)^{-1} \otimes \boldsymbol{I}_p\right)$$

where $\boldsymbol{U}_{[R]}$ are the eigenvectors associated to the Laplacian null-space of $\overline{\boldsymbol{W}}_X$. Given a graph $\boldsymbol{W}_X$, the idea is to structure the CCs' relative positions with a full-rank Gaussian model. The same model is considered for $\boldsymbol{W}_Z$, $\boldsymbol{\Theta}_Z$ and $\boldsymbol{Z}$, choosing $\nu_Z = \nu_X + p - q$ for the Wishart prior to satisfy the assumption of theorem 2. With this in place, we aim at providing a complete coupling objective, matching the pairs $(\boldsymbol{W}_X, \boldsymbol{\Theta}_X)$ and $(\boldsymbol{W}_Z, \boldsymbol{\Theta}_Z)$. The joint negative cross-entropy can be decomposed as follows:

$$\mathbb{E}_{(\boldsymbol{W}_X, \boldsymbol{\Theta}_X)|\boldsymbol{X}} \left[\log \mathbb{P}((\boldsymbol{W}_Z, \boldsymbol{\Theta}_Z) = (\boldsymbol{W}_X, \boldsymbol{\Theta}_X)|\boldsymbol{Z})\right]$$

$$= \mathbb{E}_{\boldsymbol{W}_X|\boldsymbol{X}} \left[\log \mathbb{P}(\boldsymbol{W}_Z = \boldsymbol{W}_X|\boldsymbol{Z})\right] + \tag{8}$$

$$\mathbb{E}_{(\boldsymbol{W}_X, \boldsymbol{\Theta}_X)|\boldsymbol{X}} \left[\log \mathbb{P}(\boldsymbol{\Theta}_Z = \boldsymbol{\Theta}_X|\boldsymbol{W}_Z = \boldsymbol{W}_X, \boldsymbol{Z})\right] \tag{9}$$

where (8) is the usual coupling criterion of $\boldsymbol{W}_X$ and $\boldsymbol{W}_Z$ capturing intra-CC variability while (9) is a penalty resulting from the Gaussian structure on $\mathcal{S}_M$. Constructed as such, the above objective allows a trade-of between local and global structure preservation. Following current trends in DR [21], we propose to take care of the global structure first *i.e.* focusing on (9) before (8). The difficulty of dealing with (9) lies in the hierarchical construction of the graph and the Gaussian precision (see fig. 3). We state the following result.

**Corollary 1** *Let* $\boldsymbol{W}_X \in \mathcal{S}_W$, $\boldsymbol{L} = L(\overline{\boldsymbol{W}}_X)$ *and* $\mathcal{S}_M^q = (\ker \boldsymbol{L}) \otimes \mathbb{R}^q$, *then for all* $\varepsilon > 0$, *given the above hierarchical model, the solution of the problem:*

$$\min_{\boldsymbol{Z} \in \mathcal{S}_M^q} -\mathbb{E}_{\boldsymbol{\Theta}_X|\boldsymbol{X}} \left[\log \mathbb{P}(\boldsymbol{\Theta}_Z = \boldsymbol{\Theta}_X|\boldsymbol{W}_Z = \boldsymbol{W}_X, \boldsymbol{Z})\right]$$

*is a PCA embedding of* $\boldsymbol{U}_{[:R]} \boldsymbol{U}_{[:R]}^{\top} \boldsymbol{X}$ *where* $\boldsymbol{U}_{[:R]}$ *are the CCs' membership vectors of* $\overline{\boldsymbol{W}}_X$.

**Remark 5** *Note that while (8) approximates the objective of SNE-like methods when* $\varepsilon \to 0$, *the minimizer of (9) given by corollary 1 is stable for all* $\varepsilon$.

From this observation, we propose a simple heuristic to minimize (9) that consists in computing a PCA embedding of $\mathbb{E}_{\mathbb{P}_{\mathcal{P}_X}(\cdot; \boldsymbol{K}_X)} \left[\boldsymbol{U}_{[:R]} \boldsymbol{U}_{[:R]}^{\top}\right] \boldsymbol{X}$. The distribution of the connected components of the posterior of $\boldsymbol{W}_X$ being intractable, we resort to a Monte-Carlo estimation of the above expectation. The latter procedure called *ccPCA* aims at recovering the inter-CC structure that is filtered by SNE-like methods. *ccPCA* may then be used as initialization for optimizing (8) which is done by running the DR method corresponding to the graph priors at hand (section 3.1). This second step essentially consists in refining the intra-CC structure.

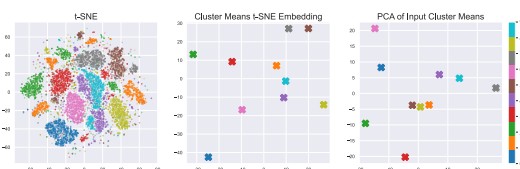

Figure 1: Left: MNIST t-SNE (perp : 30) embeddings initialized with i.i.d $\mathcal{N}(0, 1)$ coordinates. Middle: using these t-SNE embeddings, mean coordinates for each digit are represented. Right: we compute a matrix of mean input coordinates for each of the 10 digits and embed it using PCA. For t-SNE embeddings, the positions of clusters vary accross different runs and don't visually match the PCA embeddings of input mean vectors (right plot).

## 4.3 Experiments with *ccPCA*

Figure 1 shows that a t-SNE embedding of a balanced MNIST dataset of 10000 samples [14] with isotropic Gaussian initialization performs poorly in conserving the relative positions of clusters. As each digit cluster contains approximately 1000 points, with a perplexity of 30, sampling an edge across digit clusters in the graph posterior $\mathbb{P}_{\mathcal{P}_X}(\cdot; \boldsymbol{K}_X)$ is very unlikely.

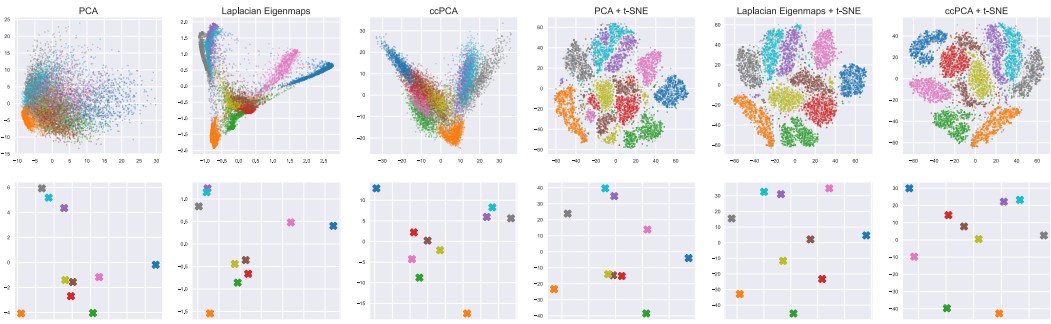

Figure 2: Top: MNIST embeddings produced by PCA, Laplacian eigenmaps, *ccPCA* and finally t-SNE launched after the previous three embeddings to improve the fine-grain structure. Bottom: mean coordinates for each digit using the embeddings of the first row. The color legend is the same as in fig. 1. t-SNE was trained during 1000 iterations using default parameters with the openTSNE implementation [34].

Recall that the perplexity value [38] corresponds to the approximate number of effective neighbors of each point. Hence images of different digits are with very high probability in different CCs of the graph posterior and their CC-wise means are not coupled as discussed in section 3.2. To remedy this in practice, PCA or Laplacian eigenmaps are usually used as initialization [21].

These strategies are tested (fig. 2) together with *ccPCA*. This shows that *ccPCA* manages to retrieve the digits that mostly support the large-scale variability as measured by the peripheral positioning of digits 0 (blue), 2 (green), 6 (pink) and 7 (grey) given by the right side of fig. 1. Other perplexity values for *ccPCA* are explored in appendix B.2 while the experimental setup is detailed in appendix B.1. In appendix B.3, we perform quantitative evaluations of *ccPCA* for both t-SNE and UMAP on various datasets using K-ary neighborhood criteria. We find that using *ccPCA* as initialization is in general more reliable than PCA and Laplacian eigenmaps for preserving global structure using both t-SNE and UMAP.

Compared to PCA, *ccPCA* manages to aggregate points into clusters, thus filtering the intra-cluster variablity and focusing solely on the inter-cluster structure. Compared to Laplacian eigenmaps which performs well at identifying clusters but suffers from the same deficiency as t-SNE for positioning them, *ccPCA* retains more of the coarse-grain structure. These observations support our unifying probabilistic framework and the theoretical results about the MRF degeneracy which are the leading contributions of this article. The *ccPCA* initialization appears as a first stepping stone towards more grounded DR methods based on the probabilistic model presented in this article.

## 5 Conclusion and Perspectives

In this work we shed a new light on most popular DR methods by showing that they can be unified within a common probabilistic model in the form of latent Markov Random Fields Graphs coupled by a cross entropy. The definition of such a model constitutes a major step towards the understanding of common dimension reduction methods, in particular their structure preservation properties as discussed in this article.

Our work offers many perspectives, among which the possibility to enrich the probabilistic model with more suited graph priors. Currently considered priors are simply the ones that are conjugate to the MRFs thus they are mostly designed to yield a tractable coupling objective. However they may not be optimal and could be modified to capture targeted features, *e.g.* communities, in the input data, and give adapted representations in the latent space. The graph coupling approach could also be extended to more general latent structures governing the joint distribution of observations. Finally, the probabilistic model could be leveraged to tackle hyper-parameter calibration, especially kernel bandwidths that have a great influence on the quality of the representations and are currently tuned using heuristics with unclear motivations.

**Acknowledgments.** The authors would like to thank the anonymous reviewers whose comments and questions helped improve the clarity of this manuscript, as well as Aurélien Garivier, Antoine Barrier and Floshi Poshi for helpful discussions. This work was supported by the Agence Nationale de la Recherche ANR-18-CE45-0023 SingleStatOmics.

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
