# A Proofs

## A.1 Proof of theorem 1

$\boldsymbol{W} \in \mathcal{S}_W$ is the weight matrix of a graph with $R$ connected components $\{C_1, ..., C_R\}$ partitioning $[n]$. Since $k$ is upper bounded by a constant, there exists $M_+ > 1$ that upper bounds $k$. Let $\boldsymbol{\mathcal{T}}$ be the adjacency matrix of a spanning forest of $\boldsymbol{W}$, since each edge of $\boldsymbol{W}$ is bounded by $n$, one has:

$$\int f_k(\boldsymbol{X}, \boldsymbol{W}) \lambda_{\mathcal{S}_C}(d\boldsymbol{X}) = \int \prod_{(i,j) \in [n]^2} k(\boldsymbol{X}_i - \boldsymbol{X}_j)^{W_{ij}} \lambda_{\mathcal{S}_C}(d\boldsymbol{X})$$

$$\leq M_+^{n^3} \int \prod_{(i,j) \in [n]^2} k(\boldsymbol{X}_i - \boldsymbol{X}_j)^{\mathcal{T}_{ij}} \lambda_{\mathcal{S}_C}(d\boldsymbol{X})$$

$$\leq M_+^{n^3} \prod_{r \in [R]} \int \prod_{(i,j) \in C_r^2} k(\boldsymbol{X}_i - \boldsymbol{X}_j)^{\mathcal{T}_{ij}} \lambda_{\mathcal{S}_C}(d\boldsymbol{X}) \, . \quad (10)$$

Let $r \in [R]$. The spanning tree corresponding to the $r^{th}$ connected component called $\boldsymbol{\mathcal{T}}^r$ has exactly $n_r - 1$ edges. There exists a leaf node $\ell \in [n]$ of $\boldsymbol{\mathcal{T}}^r$ and let $\tilde{\ell}$ be the node linked to it. Consider a bijective map $\sigma: C_r \backslash \{\ell\} \to [n_r - 1]$ such that $\sigma(\tilde{\ell}) = 1$ and for $(i,j) \in (C_r \backslash \{\ell\})^2$, $\sigma(i) \leq \sigma(j)$ implies that node $i$ has a shorter path on $\overline{\boldsymbol{\mathcal{T}}^r}$[1] to $\ell$ than node $j$. There exists a bijective map $e: [2: n_r - 1] \to [n_r - 2]$ such that for $i \in [2: n_R - 1]$, $\overline{\boldsymbol{\mathcal{T}}^r}_{\sigma^{-1}(i), \sigma^{-1}(e(i))} > 0$ and node $\sigma^{-1}(e(i))$ has a shorter path on $\overline{\boldsymbol{\mathcal{T}}^r}$ to node $\ell$ than node $\sigma^{-1}(i)$. Recall that since $\boldsymbol{X} \in \mathcal{S}_C$ one has: $\sum_{i \in C_r} \boldsymbol{X}_i = 0$ hence $\boldsymbol{X}_\ell = -\sum_{i \neq \ell} \boldsymbol{X}_i$. Let us now consider the linear map $\phi^r$ such that:

$$\forall i \in [n_r - 1], \quad \phi^r(\boldsymbol{X}_i) = \begin{cases} \boldsymbol{X}_{\sigma^{-1}(i)} + \sum_{j \in [n_r - 1]} \boldsymbol{X}_{\sigma^{-1}(j)} & \text{if } i = 1 \\ \boldsymbol{X}_{\sigma^{-1}(i)} - \boldsymbol{X}_{\sigma^{-1}(e(i))} & \text{otherwise} \, . \end{cases}$$

We now show that the change of variable $\phi^r$ is a $\mathcal{C}^1$ diffeomorphism by proving that its Jacobian has full rank. Ordering the columns with the map $\sigma$, the latter takes the form:

$$\boldsymbol{J}_{\phi^r} = \begin{pmatrix} 2 & 1 & 1 & \dots & 1 \\ & 1 & 0 & \dots & 0 \\ & & \ddots & \ddots & \vdots \\ & \boldsymbol{A} & & \ddots & 0 \\ & & & & 1 \end{pmatrix}$$

where $\boldsymbol{A}$ is a strictly lower triangular matrix such that for all $i \in [2: n_r - 1]$, $A_{ie(i)} = -1$ and for all $t \neq e(i)$, $A_{it} = 0$. The above can be factorized as:

$$\boldsymbol{J}_{\phi^r} = \begin{pmatrix} \alpha_{n_r - 1} & \alpha_{n_r - 2} & \dots & \alpha_2 & \alpha_1 \\ 0 & 1 & 0 & \dots & 0 \\ \vdots & & \ddots & \ddots & \ddots & \vdots \\ \vdots & & & \ddots & \ddots & 0 \\ 0 & & \dots & \dots & 0 & 1 \end{pmatrix}^{-1} \begin{pmatrix} 1 & 0 & \dots & \dots & 0 \\ & 1 & \ddots & & \vdots \\ & & \ddots & \ddots & \vdots \\ & \boldsymbol{A} & & \ddots & 0 \\ & & & & 1 \end{pmatrix}$$

where $\alpha_1 = -1$ and for $\ell > 1$, $\alpha_\ell = \sum_{j < l} \alpha_j \mathbb{1}_{e(n_r - j) = n_r - \ell} - 1$. With this in place, for $i \in [n_r - 1]$, $\alpha_i \neq 0$ in particular $\alpha_{n_r - 1} \neq 0$ therefore $|\boldsymbol{J}_{\phi^r}| \neq 0$ and $\phi^r$ is a $\mathcal{C}^1$ diffeomorphism. This change of variable yields:

$$\int \prod_{(i,j) \in C_r^2} k(\boldsymbol{X}_i - \boldsymbol{X}_j)^{\mathcal{T}_{ij}} \lambda_{\mathcal{S}_C}(d\boldsymbol{X}) = \int \bigotimes_{i \in [n_r - 1]} k(\boldsymbol{Y}_i) |\boldsymbol{J}_{\phi^r}(\boldsymbol{Y})|^{-1} \lambda_{\mathbb{R}^p}(d\boldsymbol{Y})$$

$$= |\boldsymbol{J}_{\phi^r}|^{-1} \prod_{i \in [n_r - 1]} \int k(\boldsymbol{Y}_i) \lambda_{\mathbb{R}^p}(d\boldsymbol{Y}_i)$$

---

[1]Symmetrized version *i.e.* $\overline{\boldsymbol{\mathcal{T}}^r} = \boldsymbol{\mathcal{T}}^r + (\boldsymbol{\mathcal{T}}^r)^\top$.

using the Fubini Tonelli theorem. The result follows from $\lambda_{\mathbb{R}^p}$-integrability of $k$ and upper bound 10.

## A.2 Proof of proposition 1

Let $\mathcal{P} \in \{B, D, E\}$, $k$ be a valid kernel (assumptions of theorem 1) with $\boldsymbol{K}_X = (k(\boldsymbol{X}_i - \boldsymbol{X}_j))_{(i,j)\in[n]^2}$ and $\boldsymbol{\pi} \in \mathbb{R}_+^{n \times n}$. Let $\boldsymbol{W} \sim \mathbb{P}_{\mathcal{P},k}^\varepsilon(\cdot\,; \boldsymbol{\pi}, 1)$. Inversion of conditional with Bayes rule gives:

$$\forall \boldsymbol{W} \in \mathcal{S}_W, \quad \mathbb{P}(\boldsymbol{W}|\boldsymbol{X}) \propto \mathcal{C}_k^\varepsilon(\boldsymbol{W})^{-1} f^\varepsilon(\boldsymbol{X}, \boldsymbol{W}) f_k(\boldsymbol{X}, \boldsymbol{W}) \mathbb{P}_{\mathcal{P},k}^\varepsilon(\boldsymbol{W}; \boldsymbol{\pi}, 1) \tag{11}$$

where the prior reads:

$$\mathbb{P}_{\mathcal{P},k}^\varepsilon(\boldsymbol{W}; \boldsymbol{\pi}, 1) \propto \mathcal{C}_k^\varepsilon(\boldsymbol{W}) \Omega_{\mathcal{P}}(\boldsymbol{W}) \prod_{(i,j)\in[n]^2} \pi_{ij}^{W_{ij}} . \tag{12}$$

Hence the joint normalizing constant simplifies such that:

$$\forall \boldsymbol{W} \in \mathcal{S}_W, \quad \mathbb{P}(\boldsymbol{W}|\boldsymbol{X}) \propto f^\varepsilon(\boldsymbol{X}, \boldsymbol{W}) \Omega_{\mathcal{P}}(\boldsymbol{W}) \prod_{(i,j)\in[n]^2} (\pi_{ij} k(\boldsymbol{X}_i - \boldsymbol{X}_j))^{W_{ij}} \tag{13}$$

$$\xrightarrow[\varepsilon \to 0]{} \Omega_{\mathcal{P}}(\boldsymbol{W}) \prod_{(i,j)\in[n]^2} (\pi_{ij} k(\boldsymbol{X}_i - \boldsymbol{X}_j))^{W_{ij}} \tag{14}$$

which ends the proof. As a complement, we now explicit the simple forms taken by the posterior limit graph in each case.

$B$-**Prior**   Recall that in this case the prior reads:

$$\mathbb{P}_B^\varepsilon(\boldsymbol{W}; \boldsymbol{\pi}, 1) \propto \mathcal{C}_k^\varepsilon(\boldsymbol{W}) \prod_{(i,j)\in[n]^2} \pi_{ij}^{W_{ij}} \mathbb{1}_{W_{ij} \le 1} .$$

Therefore the posterior limit graph has the distribution:

$$\begin{aligned}
\mathbb{P}_B(\boldsymbol{W}; \boldsymbol{\pi} \odot \boldsymbol{K}_X) &= \frac{\prod_{(i,j)\in[n]^2} (\pi_{ij} k(\boldsymbol{X}_i - \boldsymbol{X}_j))^{W_{ij}} \mathbb{1}_{W_{ij} \le 1}}{\sum_{\boldsymbol{W} \in \mathcal{S}_W} \prod_{(i,j)\in[n]^2} (\pi_{ij} k(\boldsymbol{X}_i - \boldsymbol{X}_j))^{W_{ij}} \mathbb{1}_{W_{ij} \le 1}} \\
&= \prod_{(i,j)\in[n]^2} \left( \frac{\pi_{ij} k(\boldsymbol{X}_i - \boldsymbol{X}_j)}{1 + \pi_{ij} k(\boldsymbol{X}_i - \boldsymbol{X}_j)} \right)^{W_{ij}} \left( \frac{1}{1 + \pi_{ij} k(\boldsymbol{X}_i - \boldsymbol{X}_j)} \right)^{1 - W_{ij}} \mathbb{1}_{W_{ij} \le 1} .
\end{aligned}$$

This distribution amounts to: $\forall (i,j) \in [n]^2, \quad \boldsymbol{W}_{ij} \overset{\perp\!\!\!\perp}{\sim} \mathcal{B}\left( \frac{\pi_{ij} k(\boldsymbol{X}_i - \boldsymbol{X}_j)}{1 + \pi_{ij} k(\boldsymbol{X}_i - \boldsymbol{X}_j)} \right)$.

$D$-**Prior**   The prior writes:

$$\mathbb{P}_D^\varepsilon(\boldsymbol{W}; \boldsymbol{\pi}, 1) \propto \mathcal{C}_k^\varepsilon(\boldsymbol{W}) \prod_{(i,j)\in[n]^2} \pi_{ij}^{W_{ij}} \mathbb{1}_{W_{i+}=1} .$$

The distribution of the posterior limit then becomes:

$$\begin{aligned}
\mathbb{P}_D(\boldsymbol{W}; \boldsymbol{\pi} \odot \boldsymbol{K}_X) &= \frac{\prod_{(i,j)\in[n]^2} (\pi_{ij} k(\boldsymbol{X}_i - \boldsymbol{X}_j))^{W_{ij}} \mathbb{1}_{W_{i+}=1}}{\sum_{\boldsymbol{W} \in \mathcal{S}_W} \prod_{(i,j)\in[n]^2} (\pi_{ij} k(\boldsymbol{X}_i - \boldsymbol{X}_j))^{W_{ij}} \mathbb{1}_{W_{i+}=1}} \\
&= \frac{\prod_{(i,j)\in[n]^2} (\pi_{ij} k(\boldsymbol{X}_i - \boldsymbol{X}_j))^{W_{ij}} \mathbb{1}_{W_{i+}=1}}{\prod_{i\in[n]} \sum_{\ell\in[n]} \pi_{i\ell} k(\boldsymbol{X}_i - \boldsymbol{X}_\ell)} \\
&= \prod_{(i,j)\in[n]^2} \left( \frac{\pi_{ij} k(\boldsymbol{X}_i - \boldsymbol{X}_j)}{\sum_{\ell\in[n]} \pi_{i\ell} k(\boldsymbol{X}_i - \boldsymbol{X}_\ell)} \right)^{W_{ij}} \mathbb{1}_{W_{i+}=1} .
\end{aligned}$$

This distribution amounts to: $\forall i \in [n], \quad \boldsymbol{W}_i \overset{\perp\!\!\!\perp}{\sim} \mathcal{M}\left( 1, \left( \frac{\pi_{ij} k(\boldsymbol{X}_i - \boldsymbol{X}_j)}{\sum_{\ell\in[n]} \pi_{i\ell} k(\boldsymbol{X}_i - \boldsymbol{X}_\ell)} \right)_{j\in[n]} \right)$.

*E*-**Prior**   In this case the prior reads:

$$\mathbb{P}_E^\varepsilon(\boldsymbol{W}; \boldsymbol{\pi}, 1) \propto \mathcal{C}_k^\varepsilon(\boldsymbol{W}) \prod_{(i,j)\in[n]^2} \frac{\pi_{ij}^{W_{ij}}}{W_{ij}!} \mathbb{1}_{W_{++}=n} \; .$$

Finally, deriving the distribution of the posterior graph limit:

$$\mathbb{P}_E(\boldsymbol{W}; \boldsymbol{\pi} \odot \boldsymbol{K}_X) = \frac{\prod_{(i,j)\in[n]^2}(W_{ij}!)^{-1}\left(\pi_{ij}k(\boldsymbol{X}_i - \boldsymbol{X}_j)\right)^{W_{ij}}\mathbb{1}_{W_{++}=n}}{\sum_{\boldsymbol{W}\in\mathcal{S}_W}\prod_{(i,j)\in[n]^2}(W_{ij}!)^{-1}\left(\pi_{ij}k(\boldsymbol{X}_i - \boldsymbol{X}_j)\right)^{W_{ij}}\mathbb{1}_{W_{++}=n}}$$

$$= n! \prod_{(i,j)\in[n]^2}(W_{ij})^{-1}\left(\frac{\pi_{ij}k(\boldsymbol{X}_i - \boldsymbol{X}_j)}{\sum_{(\ell,t)\in[n]^2}\pi_{\ell t}k(\boldsymbol{X}_\ell - \boldsymbol{X}_t)}\right)^{W_{ij}}\mathbb{1}_{W_{++}=n} \; .$$

This distribution amounts to: $\boldsymbol{W} \sim \mathcal{M}\left(n, \left(\frac{\pi_{ij}k(\boldsymbol{X}_i - \boldsymbol{X}_j)}{\sum_{(\ell,t)\in[n]^2}\pi_{\ell t}k(\boldsymbol{X}_\ell - \boldsymbol{X}_t)}\right)_{(i,j)\in[n]^2}\right).$

### A.3   Proof of theorem 2

We consider the following hierarchical model, for $\nu_X, \nu_Z \geq n$:

$$\boldsymbol{\Theta}_X \sim \mathcal{W}(\nu_X, \boldsymbol{I}_n)$$
$$\text{vec}(\boldsymbol{X})|\boldsymbol{\Theta}_X \sim \mathcal{N}(\boldsymbol{0}, \boldsymbol{\Theta}_X^{-1} \otimes \boldsymbol{I}_p)$$
$$\boldsymbol{\Theta}_Z \sim \mathcal{W}(\nu_Z, \boldsymbol{I}_n)$$
$$\text{vec}(\boldsymbol{Z})|\boldsymbol{\Theta}_Z \sim \mathcal{N}(\boldsymbol{0}, \boldsymbol{\Theta}_Z^{-1} \otimes \boldsymbol{I}_q) \; .$$

With this at hand, the posteriors for $\boldsymbol{\Theta}_X$ and $\boldsymbol{\Theta}_Z$ can be derived in closed form:

$$\boldsymbol{\Theta}_X|\boldsymbol{X} \sim \mathcal{W}(\nu_X + p, \left(\boldsymbol{I}_n + \boldsymbol{X}\boldsymbol{X}^\top\right)^{-1})$$
$$\boldsymbol{\Theta}_Z|\boldsymbol{Z} \sim \mathcal{W}(\nu_Z + q, \left(\boldsymbol{I}_n + \boldsymbol{Z}\boldsymbol{Z}^\top\right)^{-1}) \; .$$

Keeping terms of $-\mathbb{E}_{\boldsymbol{\Theta}_X}\left[\log \mathbb{P}(\boldsymbol{\Theta}_Z = \boldsymbol{\Theta}_X|\boldsymbol{Z})|\boldsymbol{X}\right]$ that depends on $\boldsymbol{Z}$, one has the optimization problem:

$$\min_{\boldsymbol{Z}\in\mathbb{R}^{n\times q}} \quad \frac{\nu_X + p}{2}\,\text{tr}\left(\boldsymbol{Z}^\top(\boldsymbol{I}_n + \boldsymbol{X}\boldsymbol{X}^\top)^{-1}\boldsymbol{Z}\right) - \frac{\nu_Z + q}{2}\log|\boldsymbol{I}_n + \boldsymbol{Z}\boldsymbol{Z}^\top|$$

Our strategy is to first find the optimal sample covariance matrix $\boldsymbol{Z}\boldsymbol{Z}^\top$ and then focus on the solution in $\boldsymbol{Z}$. To that extend, consider the eigendecomposition of the sample covariance matrices: $\boldsymbol{X}\boldsymbol{X}^\top = \boldsymbol{V}\boldsymbol{D}\boldsymbol{V}^\top$ and $\boldsymbol{Z}\boldsymbol{Z}^\top = \boldsymbol{U}\boldsymbol{\Lambda}\boldsymbol{U}^\top$ where $\boldsymbol{D} = \text{diag}(\boldsymbol{d})$ and $\boldsymbol{\Lambda} = \text{diag}(\boldsymbol{\lambda})$ such that $d_1 \geq ... \geq d_n$ and $\lambda_1 \geq ... \geq \lambda_n$. Denoting $\gamma = (\nu_X + q)/(\nu_Z + p)$, we consider the following problem:

$$\min_{\boldsymbol{U}\in\mathcal{O}(n), \boldsymbol{\Lambda}} \quad \text{tr}\left(\boldsymbol{U}\boldsymbol{\Lambda}\boldsymbol{U}^\top\boldsymbol{V}(\boldsymbol{I}_n + \boldsymbol{D})^{-1}\boldsymbol{V}^\top\right) - \gamma\log|\boldsymbol{I}_n + \boldsymbol{\Lambda}| \tag{15}$$

$$\text{s.t.} \quad \boldsymbol{\Lambda} \succcurlyeq \boldsymbol{0} \tag{16}$$

$$\text{rank}(\boldsymbol{\Lambda}) \leq q \tag{17}$$

The above problem is non-convex because of the rank constraint (17). Nonetheless it can be simplified as we now show.

We focus on finding the optimal eigenvectors first. To that extent, let us denote, $\boldsymbol{R} = \boldsymbol{U}^\top\boldsymbol{V}$. Only the left term in (18) depends on $\boldsymbol{R}$. The optimization problem for eigenvectors writes:

$$\min_{\boldsymbol{R}\in\mathcal{O}(n)} \quad \text{tr}\left(\boldsymbol{R}^\top\boldsymbol{\Lambda}\boldsymbol{R}(\boldsymbol{I}_n + \boldsymbol{D})^{-1}\right) \tag{18}$$

The objective (18) can be expressed as: $\sum_{(i,j)\in[n]^2}\lambda_i(1+d_j)^{-1}R_{ij}^2$. Now one can notice that since $\boldsymbol{R}$ is orthogonal, $\boldsymbol{R}\odot\boldsymbol{R}$ is doubly stochastic (*i.e.* sum of coefficients on each row and

column is equal to one). Therefore thanks to the Birkhoff–von Neumann theorem, there exists $\theta_1, ..., \theta_L \geq 0$, $\sum_{\ell \in [L]} \theta_\ell = 1$ and permutation matrices $\boldsymbol{P}_1, ..., \boldsymbol{P}_L$ such that:

$$\boldsymbol{R} \odot \boldsymbol{R} = \sum_{\ell \in [L]} \theta_\ell \boldsymbol{P}_\ell$$

where for all $\ell \in [L]$, there exists a permutation $\sigma_\ell$ of $[n]$ such that $P_{\ell, ij} = \mathbb{1}_{\sigma_\ell(i)=j}$ for $(i, j) \in [n]^2$.

With this at hand, objective (18) writes: $\sum_{\ell \in [L]} \theta_\ell \sum_{i \in [n]} \lambda_i (1 + d_{\sigma_\ell(i)})^{-1}$. There exists a permutation $\sigma^\star$ such that the quantity $\sum_{i \in [n]} \lambda_i (1 + d_{\sigma_\ell(i)})^{-1}$ is minimal. Note that the identity permutation *i.e.* for $i \in [n]$, $\sigma(i) = i$ is optimal in this case as the $(\lambda_i)_{i \in [n]}$ and the $(d_i)_{i \in [n]}$ are in decreasing order. Then choosing for $\ell \in [L]$, $\theta_\ell =_{\sigma_\ell = \sigma^\star}$ minimizes the latter quantity. Therefore the solution of (18) $\boldsymbol{R}^\star$ is such that for $(i, j) \in [n]^2$, $R^\star_{ij} = \pm \mathbb{1}_{\sigma^\star(i)=j}$. Thus an optimum in $\boldsymbol{U}$ of 18 is such that $\boldsymbol{U}^\star = \boldsymbol{V} \boldsymbol{R}^\star$.

Hence $\boldsymbol{U} = \boldsymbol{V}$, in particular, is optimal. We will choose this $\boldsymbol{U}$ in what follows as the sign of the axes do not influence the characterization of the final result in $\boldsymbol{Z}$ as a PCA embedding. Such a choice gives $\boldsymbol{Z}\boldsymbol{Z}^\top = \boldsymbol{V}\boldsymbol{\Lambda}\boldsymbol{V}^\top$.

Now it remains to find the optimal eigenvalues $(\lambda_i)_{i \in [n]}$. The rank constraint (17) can be easily dealt with: since the eigenvalues are sorted in decreasing order, the constraint implies that for $i \geq q$, $\lambda_i = 0$. Thus the eigenvalue problem can be formulated in $\mathbb{R}^q$:

$$\min_{\boldsymbol{\lambda} \in \mathbb{R}^q} \quad \boldsymbol{\lambda}^\top (\mathbf{1} + \boldsymbol{d})^{-1} - \gamma \mathbf{1}^\top \log(\mathbf{1} + \boldsymbol{\lambda}) \tag{19}$$

$$\text{s.t.} \quad \forall i \in [q], \quad \lambda_i \geq 0, \quad \lambda_1 \geq ... \geq \lambda_q \tag{20}$$

where (20) accounts for (16). The above is convex. (19) is minimized for $\boldsymbol{\lambda} = \gamma(\mathbf{1} + \boldsymbol{d}) - \mathbf{1}$. Taking the feasibility constraint (20) into account one has a solution $\boldsymbol{\lambda}^*$ such that:

$$\forall i \in [n], \quad \lambda^*_i = \begin{cases} \max(0, \gamma(1 + d_i) - 1) & \text{if} \quad i \leq q \\ 0 & \text{otherwise} . \end{cases}$$

Note that this solution is not unique if there are repeated eigenvalues. Notice also that one has the freedom to choose the Wishart prior parameters such that $\gamma = 1$. Doing so, the solution satisfies $\boldsymbol{Z}^\star \boldsymbol{Z}^{\star T} = \boldsymbol{V}_{[:,q]} \boldsymbol{D}_{[q,q]} \boldsymbol{V}^\top_{[q,:]}$. Therefore there exists $\boldsymbol{R}$ an orthogonal matrix of size $q$ such that $\boldsymbol{Z}^\star = \boldsymbol{V}_{[:,q]} \boldsymbol{D}^{\frac{1}{2}}_{[q,q]} \boldsymbol{R}$. The latter is the output of a PCA model of $\boldsymbol{X}$ with $q$ components, which is defined up to a rotation.

### A.4   Proof of Corollary 1

With the presented hierarchical model (fig. 3), the coupling problem is the following:

$$\min_{\boldsymbol{Z} \in \mathcal{S}^q_M} \quad \text{tr}\left(\boldsymbol{U}_{[:R]} \boldsymbol{Z}^\top (\boldsymbol{I}_R + \varepsilon \boldsymbol{U}^\top_{[R]} \boldsymbol{X} \boldsymbol{X}^\top \boldsymbol{U}_{[:R]})^{-1} \boldsymbol{U}^\top_{[R]} \boldsymbol{Z}\right) - \log |\boldsymbol{I}_R + \varepsilon \boldsymbol{U}^\top_{[R]} \boldsymbol{Z}\boldsymbol{Z}^\top \boldsymbol{U}_{[:R]}| \tag{21}$$

where $\boldsymbol{U}_{[:R]}$ are the eigenvectors associated to the Laplacian null-space of $\overline{\boldsymbol{W}}_X$.

Let us denote $\bar{\boldsymbol{Z}} = \boldsymbol{U}^\top_{[R]} \boldsymbol{Z} \in \mathbb{R}^{R \times q}$ and $\bar{\boldsymbol{X}} = \boldsymbol{U}^\top_{[R]} \boldsymbol{X} \in \mathbb{R}^{R \times p}$. Note that $\boldsymbol{Z} \to \boldsymbol{U}^\top_{[R]} \boldsymbol{Z}$ is a bijective linear map from $\mathcal{S}^q_M$ to $\mathbb{R}^{R \times q}$ with inverse $\bar{\boldsymbol{Z}} \to \boldsymbol{U}_{[R]} \bar{\boldsymbol{Z}}$ (and equivalently for $\mathbb{R}^{R \times p}$). Hence (21) is equivalent to:

$$\min_{\bar{\boldsymbol{Z}} \in \mathbb{R}^{R \times q}} \quad \text{tr}\left(\bar{\boldsymbol{Z}}^\top (\boldsymbol{I}_R + \varepsilon \bar{\boldsymbol{X}} \bar{\boldsymbol{X}}^\top)^{-1} \bar{\boldsymbol{Z}}\right) - \log |\boldsymbol{I}_R + \varepsilon \bar{\boldsymbol{Z}} \bar{\boldsymbol{Z}}^\top| \tag{22}$$

According to theorem 2, the solution of problem (22) is such that there exists $\boldsymbol{R}$ orthogonal, $\bar{\boldsymbol{Z}}^\star = \boldsymbol{V}_{[:,q]} \boldsymbol{S}_{[q,q]} \boldsymbol{R}$ where $\bar{\boldsymbol{X}} \bar{\boldsymbol{X}}^\top = \boldsymbol{V} \boldsymbol{S}^2 \boldsymbol{V}^\top$ is the eigendecomposition in an orthogonal basis of the among-row covariance matrix of $\bar{\boldsymbol{X}}$. Note that the solution does not depend on $\varepsilon$.

Therefore (21) is solved for $\boldsymbol{Z}^\star = \boldsymbol{U}_{[:R]} \boldsymbol{V}_{[:,q]} \boldsymbol{S}_{[q,q]} \boldsymbol{R}$. One can notice that since the singular value decomposition (*i.e.* SVD) of $\boldsymbol{U}^\top_{[R]} \boldsymbol{X}$ takes the form $\boldsymbol{V} \boldsymbol{S} \boldsymbol{B}$ where $\boldsymbol{B}$ is an semi-orthogonal matrix of size $p$, then $\boldsymbol{U}_{[:R]} \boldsymbol{U}^\top_{[R]} \boldsymbol{X} = \boldsymbol{U}_{[:R]} \boldsymbol{V} \boldsymbol{S} \boldsymbol{B}$. Noticing that $\boldsymbol{V}' = \boldsymbol{U}_{[:R]} \boldsymbol{V}$ is orthogonal, one has that $\boldsymbol{V}' \boldsymbol{S} \boldsymbol{B}$ is a compact SVD of $\boldsymbol{U}_{[:R]} \boldsymbol{U}^\top_{[R]} \boldsymbol{X}$. Therefore, since $\boldsymbol{Z}^\star = \boldsymbol{V}' \boldsymbol{S}$, $\boldsymbol{Z}^\star$ is a PCA embedding of $\boldsymbol{U}_{[:R]} \boldsymbol{U}^\top_{[R]} \boldsymbol{X}$.

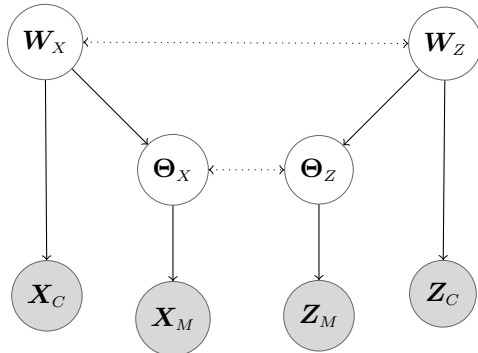

Figure 3: Graphical representation of the hierarchical model considered in section 4.2. Plain directed arrows represent conditional dependencies while dotted arrows represent the coupling links. Corollary 1 provides a solution for the coupling between $\boldsymbol{\Theta}_X$ and $\boldsymbol{\Theta}_Z$.

## B  Experiments Supplementary Material

### B.1  Experimental Setup and Details About *ccPCA*

**Implementation of existing methods.** For t-SNE, we rely on the openTSNE implementation [34] for both computing the kernel $\boldsymbol{K}_X$ with appropriate bandwidths and running the tSNE algorithm. We keep the training default parameters and 1000 iterations of gradient descent. For all experiments, the default perplexity of 30 was used to set the kernel bandwidths. For UMAP, we use the default Python implementation of [30] with default parameters. For PCA and Laplacian eigenmaps, the scikit-learn implementation is used [32] with default parameters as well.

***ccPCA.*** The pseudo code of the algorithm is given in algorithm 1. CCs' memberships (*i.e.* eigenvectors $\boldsymbol{U}_{[R]}$) are computed using igraph [13]. Regarded the time complexity of ccPCA, one can sample the posterior graph with constant time if $\mathcal{P}_X = E$, linear time if $\mathcal{P}_X = D$ and quadratic time if $\mathcal{P}_X = B$. Moreover, computing $\boldsymbol{U}_{[R]}$ can be done with linear complexity *w.r.t.* the number of nodes. Hence the time complexity is $O(N \times n)$ for $E$ and $D$ priors and $O(N \times n^2)$ for the $B$ prior, where $N$ is the number of Monte Carlo samples. In practice we found that $N \approx 100$ Monte Carlo samples produce a consistent *ccPCA* embedding for $n \approx 10000$. Note that the time complexity of PCA is $O(\min(p^3, n^3))$ where $p$ is the dimensionality (*i.e.* number of columns) of $\boldsymbol{X}$. Hence in most common applications involving images or biological sequencing data (where $p$ is very large), the additional time complexity brought by *ccPCA* compared to PCA is negligible.

---

**Algorithm 1** *ccPCA*

> **Input:** $\boldsymbol{K}_X$, $\mathcal{P}_X$, N
> **for** $\ell = 1$ **to** $N$ **do**
>     Sample $\boldsymbol{W}^\ell \sim \mathbb{P}_{\mathcal{P}_X}(\cdot; \boldsymbol{K}_X)$
>     Compute CCs' memberships $\boldsymbol{U}_{[R]}^\ell$ of $\boldsymbol{W}^\ell$
> **end for**
> **Output:** PCA of $\left( N^{-1} \sum_{\ell \in [N]} \boldsymbol{U}_{[R]}^\ell \boldsymbol{U}_{[R]}^{\ell\,T} \right) \boldsymbol{X}$

---

All experiments are performed on a machine with four Intel Core i5 processors and 16 GB memory.

### B.2  *ccPCA* with Varying Perplexity Values

Recall that the *ccPCA* algorithm retrieves the same latent graph as neighbor embedding methods. As shown in section 3.1, these graphs' distributions depend on the type of prior considered, and take simple forms as follows, when $\boldsymbol{\pi}_X = \mathbf{1}$ :

- if $\mathcal{P} = B$, $\forall (i,j) \in [n]^2$, $W_{ij} \overset{\perp}{\sim} \mathcal{B}\left(K_{X,ij}/(1 + K_{X,ij})\right)$

- if $\mathcal{P} = D$, $\forall i \in [n]$, $\boldsymbol{W}_i \overset{\perp}{\sim} \mathcal{M}\left(1, \boldsymbol{K}_{X,i}/K_{X,i+}\right)$

- if $\mathcal{P} = E$, $\boldsymbol{W} \sim \mathcal{M}\left(n, \boldsymbol{K}_X/K_{X,++}\right)$

and $\boldsymbol{K}_X$ is the kernel matrix evaluated on the data such that:

$$\forall (i,j) \in [n]^2, \quad \boldsymbol{K}_{X,ij} = k((\boldsymbol{X}_i - \boldsymbol{X}_j)/\tau_i)$$

where $\boldsymbol{\tau} \in \mathbb{R}^n$ is set using an heuristic depending on the method considered [38, 30, 36]. In fig. 4, we focus on the effect of the kernel bandwidths on *ccPCA*, choosing the example of t-SNE.

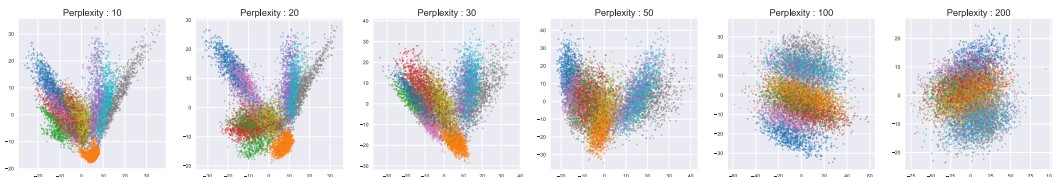

Figure 4: *ccPCA* launched for different values of the perplexity parameter. The latter determines the kernel bandwidths and can be interpreted as the number of effective neighbors of each point [38]. As the perplexity grows, the probability of connecting different clusters of digit by sampling through the graph posterior $\mathbb{P}_{\mathcal{P}_X}(\cdot; \boldsymbol{K}_X)$ increases. Therefore clusters are less and less identifiable as the perplexity increases.

From fig. 4, one can notably notice that using a high perplexity leads to a more connected graph and therefore a PCA-like embedding with less degeneracy and no clustering effect. Recall that *ccPCA* computes the same clusters as t-SNE through the CCs of the latent MRF and manage to position t-SNE clusters by focusing on their relative positions (that are filtered by t-SNE). In the case of a connected graph (high perplexity), *ccPCA* will show little advantage over classical PCA since there will not be any cluster to position. Note that this discussion can be extended to other neighbor embedding methods equivalently. Therefore, our probabilistic framework allows us to indentify which part of information is filtered by the posterior graphs with given kernel bandwidth.

### B.3 Quantitative Evaluation of *ccPCA*

For quantitative assessment of *ccPCA*, we focused on t-SNE [38] and UMAP [30] which are the most popular neighbor embedding methods. Note that for these algorithms the initialization is crucial for the global structure of the embeddings as shown in [21]. In addition to MNIST [14], we considered the datasets cifar-10, cifar-100 [22], fashion-MNIST [42] as well as the CD8+ T lymphocytes single cell RNA-seq dataset from [24].

We used the quantitative criterion of [25] to assess the quality of the embeddings. As mentionned in this paper, the use of this criterion appears as the general consensus in dimension reduction, a field in which building meaningful criteria is tedious. The criterion measures the rescaled average agreement between the K-ary neighbourhoods in the input and output spaces. It is constructed as follows.

We first define the following quantities for $(i,j) \in [n]^2$, $\rho_{ij} = |\ \{k : ||\boldsymbol{X}_i - \boldsymbol{X}_k||_2^2 < ||\boldsymbol{X}_i - \boldsymbol{X}_j||_2^2\}\ |$, $r_{ij} = |\ \{k : ||\boldsymbol{Z}_i - \boldsymbol{Z}_k||_2^2 < ||\boldsymbol{Z}_i - \boldsymbol{Z}_j||_2^2\}\ |$, $\nu_i^K = \{j : 1 \le \rho_{ij} \le K\}$ and $\gamma_i^K = \{j : 1 \le r_{ij} \le K\}$. The average K-ary neighbourhood preservation is rescaled to indicate the improvement over a random embedding such that:

$$R_n(K) = \frac{(n-1)Q_n(K) - K}{n - 1 - K} \tag{23}$$

where $Q_n(K) = \frac{1}{Kn} \sum_{i=1}^n |\ \nu_i^K \cap \gamma_i^K\ |$, $n$ is the number of data points and $K$ is the hyperparameter that adjusts the scale at which we are looking.

To focus on large-scale structure, K was chosen as either n/4 or n/2. As summarized by [21], current practice consists in using PCA or Laplacian eigenmaps as initialization for these

algorithms, thus we compare to these strategies. Results are displayed in table 2 and table 3, each entry being an average over 5 random seeds, with standard deviation displayed below each entry. Note that when not specified, tSNE and UMAP are initialized with an isotropic Gaussian variable.

These results show that using *ccPCA* is a reliable alternative to PCA and Laplacian eigenmaps for reproducing large-scale neighborhoods.

Table 2: $100 \times R_n(K)$ (23) for embeddings produced using t-SNE with various initializations.

| | tSNE | | PCA + tSNE | | LE + tSNE | | ccPCA + tSNE | |
|---|---|---|---|---|---|---|---|---|
| K of K-ary | n/4 | n/2 | n/4 | n/2 | n/4 | n/2 | n/4 | n/2 |
| MNIST | 18.7 | 7.4 | 28.4 | 21.9 | 26.7 | 18.5 | **31.3** | **28.5** |
| | ±2.2 | ±5.1 | ±0.3 | ±0.2 | ±0.7 | ±0.4 | ±0.4 | ±1.2 |
| cifar-10 | 20.3 | 16.4 | **36.9** | 41.9 | 25.8 | 24.1 | 36.4 | **43.4** |
| | ±3.2 | ±4.8 | ±0.6 | ±1.1 | ±0.6 | ±1.5 | ±0.4 | ±1.6 |
| cifar-100 | 21.6 | 18.2 | 38.1 | **47.5** | 23.3 | 26.5 | **39.6** | 43.6 |
| | ±3.6 | ±5.5 | ±0.4 | ±0.4 | ±1.5 | ±1.8 | ±0.7 | ±1.1 |
| fashion-MNIST | 27.2 | 12.3 | 36.9 | 28.5 | 32.0 | 25.1 | **41.6** | **35.7** |
| | ±4.3 | ±7.8 | ±0.1 | ±0.2 | ±0.8 | ±2.2 | ±0.9 | ±1.5 |
| Single Cell data | 25.7 | 22.4 | 37.7 | 29.0 | 28.1 | 31.5 | **40.1** | **34.6** |
| | ±4.8 | ±10.6 | ±2.7 | ±4.7 | ±1.5 | ±1.4 | ±1.7 | ±2.6 |

Table 3: $100 \times R_n(K)$ (23) for embeddings produced using UMAP with various initializations.

| | UMAP | | PCA + UMAP | | LE + UMAP | | ccPCA + UMAP | |
|---|---|---|---|---|---|---|---|---|
| K of K-ary | n/4 | n/2 | n/4 | n/2 | n/4 | n/2 | n/4 | n/2 |
| MNIST | 29.5 | 22.7 | **36.6** | 31.1 | 34.6 | 24.9 | 33.4 | **32.3** |
| | ±1.4 | ±2.2 | ±0.2 | ±0.5 | ±0.2 | ±0.7 | ±0.3 | ±0.5 |
| cifar-10 | 39.2 | 47.6 | 44.3 | **53.4** | 44.2 | 52.6 | **44.6** | 53.2 |
| | ±2.6 | ±1.1 | ±0.2 | ±0.1 | ±0.1 | ±0.2 | ±0.2 | ±0.2 |
| cifar-100 | 41.6 | 42.2 | 45.4 | 45.2 | 44.2 | 43.4 | **49.9** | **52.9** |
| | ±1.8 | ±0.9 | ±0.2 | ±0.1 | ±0.3 | ±0.1 | ±0.4 | ±0.6 |
| fashion-MNIST | 48.7 | 33.6 | 56.2 | 54.3 | 58.1 | 53.4 | **58.9** | **55.8** |
| | ±2.6 | ±9.5 | ±0.5 | ±0.6 | ±0.5 | ±0.6 | ±0.5 | ±0.3 |
| Single Cell data | 39.5 | 34.3 | 52.3 | 47.2 | **55.9** | 45.7 | 53.6 | **53.9** |
| | ±1.4 | ±6.1 | ±0.8 | ±6.9 | ±0.3 | ±0.9 | ±0.3 | ±1.3 |