# OpenReview forum: "A Probabilistic Graph Coupling View of Dimension Reduction"
_NeurIPS.cc/2022/Conference — NeurIPS 2022 Accept_

### Official Review · Reviewer_wgcv · 2022-07-08

**Rating:** 8
**Confidence:** 2
**Soundness:** 4 excellent
**Presentation:** 2 fair
**Contribution:** 4 excellent

**Summary:**

The authors define a generative probabilistic model for a dimensionality reduction (DR) problem that generalizes neighbor embedding methods such as t-SNE and UMAP. To embed the observations, X, into the embeddings, Z, their latent graph structures are coupled to be close. More specifically, the authors define conditional distributions of observations and embeddings latent graphs given associated kernel matrices on observations and embeddings respectively, and the cross-entropy between these two posteriors is minimized with respect to the embedding.

This cross-entropy objective takes various forms according to the graph priors used in observation and embedding latent graph conditional distributions; each existing neighbor embedding methods objectives corresponds to one of these forms. Since kernel matrices on observations and embeddings are shift-invariant in these methods, the relative positioning of observation clusters is not preserved in the embedding space. Then, the authors link PCA with cross-entropy objective when the Gaussian model is used as a kernel matrix which is well-defined and addresses the problem of preserving large-scale dependencies between different clusters. Furthermore, the hierarchical graph coupling is proposed to preserve the positioning of clusters, as well as their local structures, called ccPCA.

Finally, the authors perform an experiment on the MNIST dataset to demonstrate the performance of ccPCA compared to other large-scale preserving embedding methods, PCA and Laplacian eigenmaps, and the benefit of initialization of t-SNE with the embeddings from such methods.

The structure of the paper is as follows.
- The authors define a distribution P(X|W) defining the distribution of points X based on a Markov Random Field (MRF) weighted graph W. The potentials of this MRF are defined as $f(X_i,X_j) = k(X_i-X_j)^{W_{ij}}$, for a similarity kernel k.
- The authors define several options for the prior distribution P(W).
- The authors consider the posterior distribution P(W|X). They propose that a dimensionality reduction (DR) algorithm that converts X to Z can be understood to be optimizing X( P(W_X|X) , P(W_Z|Z) ), where X() is the cross entropy.
- The authors show how each of several existing DR methods can be expressed using this framework.

**Questions:**

In the experiments section, the similarity of cluster positioning from different methods with the PCA of cluster means is shown visually, however, the differences between them are not clear. Quantitative measures such as the similarity between clusters' distance matrices would show the goodness of methods in preserving the large-scale dependencies more clearly.

The extension of graph laplacian to the Wishart prior for a link to PCA is not very clear. More intuition on the mentioned key difference, Wishart distribution imposes full-rank constraint, with a graph laplacian would be helpful to better understand the connection between PCA and neighbor embedding methods.

**Strengths And Weaknesses:**

Clarity: I found the manuscript very difficult to follow. As written, the manuscript requires significant background I lack, particularly in measure theory. This is a shame because I think these results are important and thus the interested readership of this paper could be much broader than those who can current understand it. I think this could largely be alleviated with a better Outline section (see below). As is, the outline is too brief such that I wasn't able to understand the main idea until after spending hours trying to decipher the main sections. However, apart from this, all definitions and concepts are presented in a reasonable flow and all notations are clearly defined and used in the manuscript.

Originality: The paper includes a novel statistical view on dimensionality reduction approaches that explains their degeneracy, and has the potential to modify them to capture specified features and helps more reasonable hyper-parameter tuning.

Quality: The proposed theoretical perspective for neighbor embedding methods is very strong and insightful. The connection between cross-entropy objectives and SNE-based method objectives provides an insight into their differences and applications. I was not able to check the proofs.

Significance: This novel view of dimensionality reduction methods enables proposing the solutions for preserving large-scale dependencies. These DR methods are very widely used, so the significance is high. One limitation is that this view is not very intuitive, so may or may not yield insight into how to improve these methods.

Notes:

Outline: I couldn't tell what the posterior distribution was over. What kind of mathematical objects are W_X and W_Z? How are X/Z and W_X/W_Z related? There is no mention of a similarity kernel at all.

L30: Quote formatting errors.

L86: I didn't understand the X_M, X_C split.

L93: Symmetric how? k(x) = k(-x)?

L100: k is overloaded as Gaussian kernel and general symmetric function.

L102: I am assuming "integrable" on L97 refers to the fact that \int f_k (X,W) dx is finite. I am not familiar with the concept of lambda-integrable.

L109: C_k is not defined. I'm assuming this is the partition function defined by k,W. The difference between C (connected component) and \mathcal{C} (partition function) is confusing.

L153: What does the \sim + \models symbol denote?

---

> ### Author Response · Authors · 2022-07-31
> **Answer to Reviewer wgcv**
>
> We thank the reviewer for the careful reading of the manuscript, her/his assessment and relevant remarks.
>
> ### Notes
>
> We thank the reviewer for noticing typos and formatting errors. We will correct them in the revised version. Some specific answers can be found below.
>
> > L86: I didn't understand the X_M, X_C split.
>
> We thank the reviewer for pointing this imprecision. We will provide a better definition in the revised version.
> Given a graph with weights $W$, $X_M$ is the projection of $X$ onto the null-space of the graph Laplacian of this graph while $X_C$ is the projection onto the orthogonal complement to that space. Therefore $X_M$ is the mean of $X$ over the connected components of $W$ and $X_C$ is centered on these connected components. As we show, for neighbor embedding methods $X_M$ has infinite variance (diffuse distribution) therefore information about the positions of clusters (or equivalently the connected components of the graph) is lost.
>
> > L93: Symmetric how? k(x) = k(-x)?
>
>  The reviewer is correct.
>
> > L102: I am assuming "integrable" on L97 refers to the fact that \int f_k (X,W) dx is finite. I am not familiar with the concept of lambda-integrable.
>
> Throughout the paper, lambda is the Lebesgue measure hence the reviewer is absolutely correct.
>
> > L109: C_k is not defined. I'm assuming this is the partition function defined by k,W. The difference between C (connected component) and \mathcal{C} (partition function) is confusing.
>
> $C_k$ is indeed the partition function of the MRF. Notice that when k is the Gaussian kernel $C_k$ is the pseudo-determinant of the graph Laplacian $L$, where the pseudo-determinant is defined as the product of the non-null eigenvalues ($L$ is always positive semi-definite). We will add details about this in the revised version.
>
> > L153: What does the \sim + \models symbol denote?
>
> It means that the variables are independent. We will state it more clearly in the future version.
>
> ### Questions
>
> > In the experiments section, the similarity of cluster positioning from different methods with the PCA of cluster means is shown visually, however, the differences between them are not clear. Quantitative measures such as the similarity between clusters' distance matrices would show the goodness of methods in preserving the large-scale dependencies more clearly.
>
> We provide such quantitative criteria on a variety of datasets in appendix B.3. These results will be highlighted in the revised version.
>
> > The extension of graph laplacian to the Wishart prior for a link to PCA is not very clear. More intuition on the mentioned key difference, Wishart distribution imposes full-rank constraint, with a graph laplacian would be helpful to better understand the connection between PCA and neighbor embedding methods.
>
> We thank the reviewer for giving us the occasion to clarify this central point. Let us consider the Gaussian kernel for SNE-like methods for simplicity. Then  as we showed line 100 the conditional distribution, for SNE-like methods, takes the form:
> $$X | W \sim \mathcal{N}(0, L^{-1} \otimes I) \quad (1)$$
> where $L$ is the graph Laplacian of the discrete graph $W$.
> For PCA, one has for a positive definite matrix $\Theta_X$:
> $$X | \Theta_X \sim \mathcal{N}(0, \Theta_X^{-1} \otimes I) \quad (2)$$
> Therefore for SNE-like methods it is the graph Laplacian of $W$ that plays the role of precision among rows whereas for PCA it is a positive definite matrix (nonsingular). In both cases, in order to retrieve a tractable posterior, we select a prior that is conjugate with the conditional. For PCA, the Wishart prior for $\Theta_X$ is conjugate with $(2)$ whereas for SNE-like methods, it is the distribution given by definition 2 which is conjugate with $(1)$ (note that the prior we construct for $W$ amounts to a prior on $L$). Note that when using Gaussian kernel, the latter is similar to a Wishart distribution as $C_k$ is the pseudo determinant of $L$.
> The central difference is that $(1)$ is degenerate whereas $(2)$ is a *proper distribution* that is integrable on the whole definition space of $X$. The degeneracy of $(1)$ is an invariance with respect to the mean coordinates of $X$ on the CCs of $W$ (i.e. $X_M$) , as $\text{ker}(L)$ is spanned by the CCs indicator vectors. Therefore it results in a loss of global structure (see sections 3.2 and 4 for more details).

---

### Official Review · Reviewer_V1Te · 2022-07-10

**Rating:** 4
**Confidence:** 4
**Soundness:** 3 good
**Presentation:** 3 good
**Contribution:** 2 fair

**Summary:**

Authors of this paper introduce a unifying statistical framework based on coupling of hidden graphs through pairwise MRF using cross entropy which explains several existing dimension reduction methods with respect to different graph priors. With the above framework, authors further propose to mitigate the global structure deficiency with a new initialization method called ccPCA to existing SNE-like methods.

**Questions:**

In many places of the manuscript, authors mentioned the assumptions of theorem 1. It will be better to describe these assumptions in detail and clarify in what situations these assumptions can be satisfied.

It is interesting to see a unified probabilistic framework to cover SNE-like methods associated to different graph priors shown in section 3.1. However, the most attracting point might be even better to use the identified differences (graph priors) to analyze the pros and cons of SNE-like methods. This will provide valuable guidance for the users of SNE-like methods. Unfortunately, this type of discussion is missing.

Another direction of the unified framework is to inspire new models. Authors did this in section 4 to enrich SNE-like method by faithfully representing global structure in low dimensional space. It is expected that the new model can be addressed, but a heuristic two-step approach is used. Although the connection between PCA and the subproblem of the new model is presented, additional concerns come up too like inefficiency of Monte-Carlo estimation for W_x and suboptimality of two-step approach with early stopping as stated in the caption of Figure 2.

The experiments in section 4.3 can be improved significantly. The visual results shown on MNIST dataset are rather subjective without the true positions of different digits. Authors might conduct experiments on some simulated data with true global structure to valid the hypothesis. Moreover, the mean coordinates can be less accurate to reflect the shape of clusters due to the nonlinearity of SNE-like methods. A better way to demonstrate the differences between compared methods is preferred.


**Limitations:**

Yes

**Strengths And Weaknesses:**

Strengths:

1.	The shift-invariant pairwise MRF is used to model the dependency of observations, which leads to the unifying cross entropy losses with different graph priors.

2.	Various well-known and popularly used DR methods are recovered by the proposed framework.

3.	A new initialization method ccPCA to existing SNE-like methods is inspired by hierarchical graph coupling.

Weaknesses:

1.	The discussion on the pros and cons of existing SNE-like methods covered by the proposed framework with respect to graph priors is missing.

2.	The new model motivated by the framework for balancing inter and intra structures are handled by a heuristic approach.

3.	Visual experiments with ccPCA focusing on the relative positioning of different digits using MNIST are subjective.

---

> ### Author Response · Authors · 2022-07-31
> **Answer to Reviewer V1Te**
>
> We thank the reviewer for the insightful comments. Answers to the questions raised may be found below.
>
> ### First question
>
> As we will clarify in the revised version, the assumptions are that for a shift-invariant kernel of the form $k(x-y)$, $k$ needs to be a bounded and integrable function. These assumptions are very mild. To the best of our knowledge, every shift-invariant kernel that  are used in a dimension reduction setting satisfy this assumption. Notably, heavy-tail kernels like the student-t kernel are included. The result of theorem 1 is strong in that sense as heavy-tail kernels are now ubiquitous in dimension reduction methods.
>
> ### Second question
>
> The reviewer is raising an important question which is the comparison and understanding of neighbor embedding methods based on the distributions of the latent graphs. In the present work, the distributions of the latent graphs are leveraged to identify which structure in the input dataset $X$ is lost, which is the positions of the mean coordinates on the connected components of the latent graphs. The latter depends on the choice of prior which is specific to each method, therefore we believe our work can help practitioners in deciphering the empirical performance of each algorithm. For an experimental study of the effect of each distributions on the embeddings, we refer to the rich literature about comparative experimental studies in dimension reduction, as each method has been extensively benchmarked.
>
> In our future work, we will focus on studying which discrete graph prior is more suited to each specific use case (for instance identifying communities) as we believe there is not a unique best graph prior for dimension reduction. This echoes our conclusion about enriching the probabilistic model with more suited graph priors. In the present work, after presenting the unifying framework, we chose to focus on large scale structure and uncovered a new link with PCA. Indeed the latter naturally appears as an extension of graph coupling in which the MRF is not degenerate.
>
> ### Third question
>
> We understand the reviewer's concerns about the two-step approach. We would like to argue that dealing with global structure at initialization is typically what is done in practice when, for instance, initializing t-SNE with PCA and UMAP with Laplacian eigenmaps, see e.g. [1]. Practitioners seem to value the ability of t-SNE and UMAP to reproduce small scale dependencies and clusters. Hence methods that modify the learning objectives to take into account global structure appear less popular in practice, see for instance [2] and [3]. We refer to [5] for details on how practitioners use neighbor embedding methods in practice. Hence the most popular way to improve global structure for neighbor embedding algorithms is to improve initialization. Therefore our strategy was to cope with this practice. Note that unlike naive initialization with PCA or Laplacian eigenmaps, ccPCA is built to target specifically the information that is left unstructured by neighbor embeddings algorithms, that is the relative positions of the clusters.
>
> We discuss the time complexity of Monte Carlo estimation in ccPCA from line 570 to 580 in the appendix. As pointed out, the additional computational complexity brought by ccPCA compared to PCA is negligible.
>
> ### Fourth question
>
> Quantitative evaluation of ccPCA is performed in section B.3 of the appendix. We experiment on various popular datasets including single cell sequencing data. Both t-SNE and UMAP are tested. We measure the average agreement between the K-ary neighborhoods in the input and embedded spaces, with large K to focus on global structure. Such criterion is notably used in [4]. Note that this criterion does not depend on the mean coordinates as it does not consider labeled clusters. Some of these results will be added in the extra page of the main paper if our submission is accepted.
>
> About section 4.3, note that information about *the true positions of different digits* is provided by the right side of figure 1, where we embedded the 10 high-dimensional digit means using PCA. We thank the reviewer for appropriately pointing out that this was unclear.
>
> #### References
>
> [1] Kobak, D. et al. (2021). Initialization is critical for preserving global data structure in both t-SNE and UMAP. Nature biotechnology.
>
> [2] Lee, J. A. et al. (2013). Type 1 and 2 mixtures of Kullback–Leibler divergences as cost functions in dimensionality reduction based on similarity preservation. Neurocomputing.
>
> [3] Venna, J. et al. (2010). Information retrieval perspective to nonlinear dimensionality reduction for data visualization. Journal of Machine Learning Research.
>
> [4] Lee, J. A. et al. (2015). Multi-scale similarities in stochastic neighbour embedding: Reducing dimensionality while preserving both local and global structure. Neurocomputing.
>
> [5] Kobak, D. et al. The art of using t-sne for single-cell transcriptomics. Nature communications.

---

### Official Review · Reviewer_QqSP · 2022-07-11

**Rating:** 5
**Confidence:** 3
**Soundness:** 3 good
**Presentation:** 3 good
**Contribution:** 3 good

**Summary:**

This paper introduces a new dimensionality reduction technique by formulating a generative probabilistic model that can be used to describe popular techniques such as t-SNE, UMAP, etc. Specifically by, showing that they are all graph coupling with different priors for discrete latent structuring graphs. The authors present theoretical results describing the behavior of such models and also empirical results with a newly proposed initialization method.

**Questions:**

* In the abstract the below statement is made. Why are probabilistic foundations required for such a full understanding? These approaches such as UMAP do have foundations with technical substance as I understand. I assume what is meant is that the kind of limitations and properties presented in this paper are not possible with with UMAP / t-SNE presentation as is.
> Though widely used, these approaches lack clear probabilistic foundations to enable a full understanding of their properties and limitations.

**Limitations:**

Yes

**Strengths And Weaknesses:**

This paper presents an interesting view on popular dimensionality reduction techniques. Among its strengths I believe are:

* Lays foundation for analyzing and understanding dimensionality reduction with MRF lens. Shows interesting analytical insights on existing works.
* Provides recommendations for new ways to initialize, ccPCA, which has empirical advantages in the experiments

Weaknesses of the paper include:
* Perhaps I am misguided, but when I think about dimensionality reduction, i think most about its empirical benefits and use cases. I would have expected and liked the paper to describe more about technical details of experiments and run experiments that give the reader a full sense of what kinds of datasets certain methods work well on and on what datasets they do not.
* This could be achieved by describing properties of the datasets in ways that a practitioner could analyze about their own data. I'm also greatly curious about things like hyperparameter sensitivity and tuning, more about efficiency, scaling and the like.
* I'm also thirsting to understand more about whether the improved performance can be traced back to parts of the improved initialization. I.e. how does initialization quality relate to end quality for points. Apologies if I have missed this.

Typos:
30 - Quotation marks
146 - edges -> edge's

Possible related work:
Bayesian Distance Clustering. Leo L Duan, David B Dunson. https://arxiv.org/abs/1810.08537

---

> ### Author Response · Authors · 2022-07-30
> **Answer to Reviewer QqSP**
>
> We would like to thank the reviewer for his/her careful reading of the manuscript and insightful comments.
>
> > Perhaps I am misguided, but when I think about dimensionality reduction, i think most about its empirical benefits and use cases. I would have expected and liked the paper to describe more about technical details of experiments and run experiments that give the reader a full sense of what kinds of datasets certain methods work well on and on what datasets they do not.
>
> > This could be achieved by describing properties of the datasets in ways that a practitioner could analyze about their own data. I'm also greatly curious about things like hyperparameter sensitivity and tuning, more about efficiency, scaling and the like.
>
> The methods we study have been around for quite a long time and used extensively in practice. Empirical evaluations of these methods are at the heart of many papers. Among many others, one has for instance: [1], [2] and [3]. Our work differs in that it is mostly a theoretical contribution. Indeed the main goal of our paper is to build a unifying probabilistic framework for existing dimension reduction algorithms. Therefore we refer to the litterature for questions about hyperparameter tuning and complexity of the existing methods. We thank the reviewer for giving us the occasion to clarify this point.
>
> > I'm also thirsting to understand more about whether the improved performance can be traced back to parts of the improved initialization. I.e. how does initialization quality relate to end quality for points. Apologies if I have missed this.
>
> When it comes to neighbor embedding methods, it is well known that the relative positions of clusters mainly depends on initialization. To get a glimpse of this effect, on can for instance look at figure 2 in our paper. The figure shows that the final relative positions of clusters is determined by initial relative positions, even though the shape of the point cloud changes. Among other work, we refer to [4] for in-depth study of the importance of initialization. Our work uncovers an explanation of this fact, as we show that neighbor embedding objectives do not take into account the positions of CCs due to the degeneracy of the MRF with shift invariant kernels.
>
> > In the abstract the below statement is made. Why are probabilistic foundations required for such a full understanding? These approaches such as UMAP do have foundations with technical substance as I understand. I assume what is meant is that the kind of limitations and properties presented in this paper are not possible with with UMAP / t-SNE presentation as is.
>
> The reviewer is indeed correct. This sentence should be rephrased accordingly in the revised version. We would like to point out that UMAP's foundations are mainly rooted in algebraic topology. Our contribution provides a unifying framework based on probabilistic modeling and graphical models. As such, we believe that our work could help the machine learning community in making sense of these methods.
>
> To the best of our knowledge, it is the first time such a link is uncovered between neighbor embedding methods. We even go further and establish a new connexion with PCA, which we believe can be extended to spectral methods in general (see for instance the discussion with reviewer 54Sc).
>
> #### References
>
> [1] Dmitry Kobak and Philipp Berens. The art of using t-sne for single-cell transcriptomics. Nature communications, 10(1):1–14, 2019.
>
> [2] Huang, H., Wang, Y., Rudin, C., & Browne, E. P. (2022). Towards a comprehensive evaluation of dimension reduction methods for transcriptomic data visualization. Communications Biology, 5(1), 1-11.
>
> [3] Yingfan Wang, Haiyang Huang, Cynthia Rudin, and Yaron Shaposhnik. Understanding how dimension reduction tools work: an empirical approach to deciphering t-sne, umap, trimap, and pacmap for data visualization. J Mach. Learn. Res, 22:1–73, 2021.
>
> [4] Kobak, D., & Linderman, G. C. (2021). Initialization is critical for preserving global data structure in both t-SNE and UMAP. Nature biotechnology, 39(2), 156-157.

---

> > ### Comment · Reviewer_QqSP · 2022-08-08
> > **Thank you for your response**
> >
> > Apologies for entering this discussion so late. Thank you so much for your time and effort responding to my concerns. This is quite helpful to my understanding.

---

### Official Review · Reviewer_54Sc · 2022-07-18

**Rating:** 6
**Confidence:** 2
**Soundness:** 3 good
**Presentation:** 2 fair
**Contribution:** 3 good

**Summary:**

The paper proposes a novel principled view of dimensionality reduction using probabilistic graph coupling that allows to retrieve several modern approaches such as SNE, t-SNE, UMAP and LargeVis, and more classical ones such as PCA and its kernelized variants. The
former list of methods performs exceptionally well in identifying local dependency structure among variables at the expense of losing global structural information, identifying which has been more suited to PCA-based approaches so far.

The main theoretical contributions of the paper are:
1) Identifying the previously mentioned approaches as special cases of a larger unifying statistical framework that is formulated using probabilistic graph coupling;
2) Providing a theoretical explanation for the unsatisfactory performance at capturing global structure of the previous neighbour embedding methods;
3) Proposing a hierachical extension for neighbour embeddings, which allows to better represent the inter-cluster structure effectively leading to a novel regularized loss function, which contains an additional penalty term with respect to the global structure. In practice, the two cross-entropy terms corresponding to the local / global structure are minimized separately with solving for the global structure first, hence this effectively leads to a new initialization called ccPCA for neighbour embeddings.

Experiments are conducted on MNIST to demonstrate the previously mentioned drawbacks of these methods, that demonstrate the value of the new initialization procedure with further experiments reported in the experiments that overall demonstrate the usefulness of the proposed initialization.

**Questions:**

Suggestions:
- Some notation was left undefined (e.g. $\otimes$, $\mathcal{S}_{++}^n$, on page 7 $\mathcal{W}$) , while others were defined in a cluttered manner. My suggestion would be to have a notation paragraph at the beginning of Section 2 that readers can refer to.
- It is unfortunate that the bulk of the experiments are deferred to the Appendix, as the empirical contribution could also be useful for practitioners. I think, at least for me, it would help the reading experience to move some non-central parts of the theoretical exposition into the appendix to ease the exposition in lieu of further experiments.

Questions:
- The authors mentioned that kernelized variants of PCA can also be recovered as part of the framework. Is this something that's implicitly included in the current theory, or is this a generalization that is left to future work?
- In the conclusion it is mentioned that the proposed framework could support hyperparameter learning for the proposed approaches, such as the bandwidth for kernel based methods. This could be quite useful as the current heuristic approaches for unsupervised learning can be quite lacking. Is this something the authors have experimented with, or if any suggested approaches could be mentioned to highlight some future work direction?


**Limitations:**

Yes.

**Strengths And Weaknesses:**

This is a very interesting theoretical paper, which sheds new light on existing dimensionality reduction techniques. The paper does a good job at citing related work in detail, and the presentation of the theory is nicely structured. Maybe one drawback is that the theory is very dense in maths, and at the same time, the paper draws from a wide range of areas (which is a benefit of course), but this makes the formulation quite difficult to follow until the end, while the experiments are mostly squeezed into the last page / appendix. Overall the paper could be interesting for a wide range of audience due to the overarching popularity of the investigated methods, but the presentation may limit the accessibility. On the experiments' side, the main empirical contribution is investigating the new theoretically motivated initialization scheme, which shows superior performance compared to previous heuristic approaches.

---

> ### Author Response · Authors · 2022-07-30
> **Answer to Reviewer 54Sc**
>
> We thank the reviewer for her/his insightful comments. We will consider them and update our manuscript accordingly. Below are the answers to the reviewer's questions.
>
> > The authors mentioned that kernelized variants of PCA can also be recovered as part of the framework. Is this something that's implicitly included in the current theory, or is this a generalization that is left to future work?
>
> This generalization is left for future work. We can nonetheless give a brief overview of the result. Consider that the feature map is Gaussian: $\phi \sim \mathcal{N}(0, \Theta_X^{-1} \otimes I)$ (future work should focus on studying what assumptions are required to satisfy this). Then keeping the same model for $Z$ that is $Z \sim \mathcal{N}(0, \Theta_Z^{-1} \otimes I)$ and the Wishart priors for both $\Theta_X$ and $\Theta_Z$, applying a similar proof as in theorem 2 would lead to diagonalizing $\phi \phi^\top$ which is the kernel matrix. Hence the solution in $Z$ of the graph coupling problem would be a kernel PCA embedding.
>
> > In the conclusion it is mentioned that the proposed framework could support hyperparameter learning for the proposed approaches, such as the bandwidth for kernel based methods. This could be quite useful as the current heuristic approaches for unsupervised learning can be quite lacking. Is this something the authors have experimented with, or if any suggested approaches could be mentioned to highlight some future work direction?
>
> This problem is indeed of primary importance for neighbor embedding methods. We believe the absence of grounded methods for kernel bandwidths is due to the absence of a proper probabilistic model. As we provide one in our work, strategies like maximum likelihood estimation are now feasible. Recall that for all neighbor embedding methods a Gaussian kernel is used in the space of the input $X$. Therefore as we show, having a bandwidth parameter $\Gamma = \text{diag}(\gamma_i)\_{i \in [n]}$ boils down to the conditional
> $\mathbb{P}( X | W, \Gamma) \propto \exp \left( -\frac{1}{2} \sum\_{i,j} \gamma_i W\_{ij} || X\_i - X\_j ||\_2^2 \right)$.
> Note that the latter is equivalent to $X| W, \Gamma \sim \mathcal{N}(0, \tilde{L} \otimes I)$ where $\tilde{L}$ is the graph Laplacian of $\Gamma W + W^\top \Gamma$. With this at hand, one could focus on solving $\max_{\Gamma} \mathbb{P}(X | \Gamma)$. Note that marginalization over $W$ is done by leveraging the topological constraints imposed (such as outdegree one structure). One can even think of an empirical Bayes strategy that would consist in solving $\max_{\Gamma} \mathbb{P}(\Gamma | X)$ after choosing a suitable prior for $\Gamma$. In this case one can consider independent Gamma prior for $\gamma$ since it is conjugate with the Gaussian conditional $\mathcal{N}(0, \tilde{L} \otimes I)$. In this setting it would for instance be interesting to relate the perplexity parameter to the parameters of the Gamma prior. These ideas are examples of how our work can be leveraged to build more statistically grounded methods for hyperparameter calibration.

---

### Meta-Review · Area_Chair_xqkA · 2022-08-27

**Recommendation:** Accept
**Confidence:** Less certain

**Metareview:**

This paper aims to develop a statistical framework for developing a more rigorous understanding of well known methods such as t-sne and UMAP. There was consensus among the reviewers that the proposed framework makes considerable progress in understanding the mentioned methods.

**Award:**

No

---

### Decision · Program_Chairs · 2022-09-14

Accept